# Alterations of specific cortical GABAergic circuits underlie abnormal network activity in a mouse model of Down syndrome

Javier Zorrilla de San Martin*, Cristina Donato†, Jérémy Peixoto‡, Andrea Aguirre, Vikash Choudhary, Angela Michela De Stasi, Joana Lourenço, Marie-Claude Potier*, Alberto Bacci*

Institut du Cerveau (ICM), CNRS UMR 7225 – Inserm U1127, Sorbonne Université, Paris, France

**Abstract** Down syndrome (DS) results in various degrees of cognitive deficits. In DS mouse models, recovery of behavioral and neurophysiological deficits using GABA$_A$R antagonists led to hypothesize an excessive activity of inhibitory circuits in this condition. Nonetheless, whether over-inhibition is present in DS and whether this is due to specific alterations of distinct GABAergic circuits is unknown. In the prefrontal cortex of Ts65Dn mice (a well-established DS model), we found that the dendritic synaptic inhibitory loop formed by somatostatin-positive Martinotti cells (MCs) and pyramidal neurons (PNs) was strongly enhanced, with no alteration in their excitability. Conversely, perisomatic inhibition from parvalbumin-positive (PV) interneurons was unaltered, but PV cells of DS mice lost their classical fast-spiking phenotype and exhibited increased excitability. These microcircuit alterations resulted in reduced pyramidal-neuron firing and increased phase locking to cognitive-relevant network oscillations in vivo. These results define important synaptic and circuit mechanisms underlying cognitive dysfunctions in DS.

*For correspondence:
javier.zorrilla@icm-institute.org
(JZSM);
marie-claude.potier@upmc.fr (M-CP);
alberto.bacci@icm-institute.org
(AB)

Present address: †Luxembourg Centre for Systems Biomedicine, University of Luxembourg, Belvaux, Luxembourg; ‡Institut Pasteur, Paris, France

Competing interests: The authors declare that no competing interests exist.

## Introduction

Down syndrome (DS) is a condition caused by full or partial trisomy of human chromosome 21, characterized by various physical and neurological features including mild to severe intellectual disability (*Antonarakis et al., 2020*). Individuals with DS present important deficits in cognitive tasks known to depend on the anatomical and functional integrity of the frontal lobe (*Lee et al., 2015*). Moreover, DS is also associated with other CNS-mediated phenotypes, including an ultra-high risk for developing Alzheimer's disease and high rates of autism for which the mechanisms are unknown (*DiGuiseppi et al., 2010*; *Wiseman et al., 2015*). Interventions to ameliorate DS-mediated cognitive dysfunctions are limited. The development of interventions for this vulnerable group of individuals can be achieved through a better understanding of the mechanisms underlying a core feature of DS, such as intellectual disability.

Importantly, mouse models of DS recapitulate several cognitive deficits of this condition (*Herault et al., 2017*; *Olmos-Serrano et al., 2016b*). One of the best-characterized mouse models is the Ts65Dn mouse line (herein referred to as Ts), which carries a partial trisomy of a segment of the mouse chromosome 16 (*Davisson et al., 1993*). Ts mice recapitulate several dysfunctions present in DS individuals, such as reduced birthweight, male sterility, abnormal facial appearance and several cognitive impairments, including executive functions, such as working memory and cognitive flexibility (*Olmos-Serrano et al., 2016b*).

**eLife digest** Down syndrome is a genetic disorder caused by the presence of a third copy of chromosome 21. Affected individuals show delayed growth, characteristic facial features, altered brain development; with mild to severe intellectual disability. The exact mechanisms underlying the intellectual disability in Down syndrome are unclear, although studies in mice have provided clues. Drugs that reduce the inhibitory activity in the brain improve cognition in a mouse model of Down syndrome. This suggests that excessive inhibitory activity may contribute to the cognitive impairments.

Many different neural circuits generate inhibitory activity in the brain. These circuits contain cells called interneurons. Sub-types of interneurons act via different mechanisms to reduce the activity of neurons. Identifying the interneurons that are affected in Down syndrome would thus improve our understanding of the brain basis of the disorder.

Zorrilla de San Martin et al. compared mice with Down syndrome to unaffected control mice. The results revealed an increased activity in two types of inhibitory brain circuits in Down syndrome. The first contains interneurons called Martinotti cells. These help the brain to combine inputs from different sources. The second contains interneurons called parvalbumin-positive basket cells. These help different areas of the brain to synchronize their activity, which in turn makes it easier for those areas to exchange information.

By mapping the changes in inhibitory circuits in Down syndrome, Zorrilla de San Martin et al. have provided new insights into the biological basis of the disorder. Future studies should examine whether targeting specific circuits with pharmacological treatments could ultimately help reduce the associated impairments.

Recovery of behavioral and neurophysiological deficits underlying cognitive impairments using GABA$_A$ receptor blockers led to the hypothesis that intellectual deficits in DS are produced by an excessive activity of inhibitory circuits (*Fernandez et al., 2007*; *Zorrilla de San Martin et al., 2018*). Nonetheless, direct evidence for over-inhibition in DS is lacking. Moreover, given the anatomical, molecular and functional diversity of cortical inhibitory neurons (*Tremblay et al., 2016*), the functional implications of this hypothesis at the network level, as well as the involvement of specific GABAergic circuits remain obscure.

Executive functions depend on the integrity of the prefrontal cortex (PFC), which plays an essential role in the synchronization of task-relevant, large-scale neuronal activity (*Helfrich and Knight, 2016*). An important network correlate of this synchronization is represented by neuronal oscillations: rhythmic fluctuations of the electrical activity of single neurons, local neuronal populations and multiple neuronal assemblies, distributed across different brain regions (*Buzsáki and Wang, 2012*). Oscillations are the result of a balanced and coordinated activity of excitatory pyramidal neurons (PNs) and a rich diversity of inhibitory neurons that use γ-aminobutiric acid (GABA) as neurotransmitter. In particular, parvalbumin (PV)-positive inhibitory interneurons form synapses onto the perisomatic region of PNs. PV cells thus tightly control PN spiking activity and drive network oscillations in the γ-frequency range (30–100 Hz) (*Buzsáki and Wang, 2012*). γ-Oscillations are necessary for several PFC cognitive functions, such as sustained attention (*Kim et al., 2016b*) and cognitive flexibility (*Cho et al., 2015*). Conversely, Martinotti cells (MCs) are somatostatin (SST)-positive interneurons that inhibit distal dendrites of PNs, thereby controlling the integration of distal dendritic glutamatergic synaptic inputs originating from different regions of the brain (*Tremblay et al., 2016*). Dendritic integration of multi-pathway inputs is necessary for working memory (*Abbas et al., 2018*; *Kim et al., 2016a*). Therefore, PV interneurons and MCs represent two major cortical inhibitory circuits, characterized by a precise division of labor during cortical activity. Both forms of inhibition were shown to be involved in the entrainment of network oscillations (*Cardin et al., 2009*; *Chen et al., 2017*; *Sohal et al., 2009*; *Veit et al., 2017*) and in the cognitive performance during medial (m)PFC-dependent tasks (*Abbas et al., 2018*; *Cho et al., 2015*; *Cummings and Clem, 2020*). In particular, inhibition from SST interneurons plays a crucial role in mPFC-dependent memory (*Abbas et al., 2018*; *Cummings and Clem, 2020*).

Broad-spectrum GABA$_A$R antagonists are not clinically viable, as they can yield undesired seizure-like activity and/or anxiety. Interestingly, however, treatment of Ts mice with selective and partial negative allosteric modulators of α5-containing GABA$_A$Rs (α5 inverse agonist or α5IA) reverse cognitive behavioral and long-term synaptic plasticity deficits in DS mice (*Braudeau et al., 2011*; *Duchon et al., 2020*; *Martínez-Cué et al., 2013*; *Schulz et al., 2019*). Importantly, neocortical dendritic synaptic inhibition of PNs from MCs relies on α5-containing GABA$_A$Rs (*Ali and Thomson, 2008*). The preference for this specific GABA$_A$R subunit was also recently demonstrated at the equivalent hippocampal dendritic inhibitory circuit (*Schulz et al., 2019*; *Schulz et al., 2018*), raising the question of whether dendritic inhibition is specifically altered in DS.

Here we found that the dendritic synaptic inhibitory loop formed by MCs and PNs was strongly potentiated in Ts mice, with no alteration of either cell-type excitability. Conversely, the perisomatic synaptic inhibitory loop from PV cells onto PN cell bodies was unaffected in Ts mice. Strikingly, however, PV-cell excitability was strongly altered: these interneurons did not display their typical fast-spiking behavior and exhibited enhanced excitability. At the network level in vivo, these inhibitory microcircuit-specific alterations resulted in significant reduction of putative PN firing, which in turn was more tuned to β- and low γ-oscillations (10–60 Hz). These results confirm over-inhibition in DS, and reveal unexpected functional alterations of specific GABAergic circuits in this condition.

## Results

### Synaptic enhancement of dendritic inhibition in DS

Cognitive and synaptic plasticity deficits in Ts mice can be successfully treated by systemic application of a selective negative allosteric modulator of α5-containing GABA$_A$Rs, α5IA (*Braudeau et al., 2011*; *Duchon et al., 2020*; *Martínez-Cué et al., 2013*; *Schulz et al., 2019*). α5-GABA$_A$Rs are expressed at PN synapses originating from dendrite-targeting interneurons: MCs in the neocortex (*Ali and Thomson, 2008*) and O-LM in the hippocampus (*Schulz et al., 2018*). We therefore tested whether dendritic inhibition of PNs by MCs are affected in Ts mice.

We crossed Ts65Dn with GFP-X98 mice, which in the barrel cortex were shown to bias GFP expression in MCs (*Ma et al., 2006*). Accordingly, in the mPFC of these mice, GFP was expressed by a subset of SST-positive interneurons (*Figure 1—figure supplement 1a*), exhibiting a widely branched axonal plexus in L1, characteristic of dendrite-targeting inhibitory MCs (*Figure 1—figure supplement 1b,c*). Using dual whole-cell patch-clamp recordings in acute mPFC slices, we isolated unitary inhibitory postsynaptic currents (uIPSCs) in MC-PN connected pairs (*Figure 1a*). We found that dendritic MC-PN synaptic inhibition relied on α5-containing GABA$_A$Rs in both Ts and control, euploid (Eu) mice. Indeed, in both genotypes, bath application of the selective negative allosteric modulator of α5-containing GABA$_A$Rs, α5IA (100 nM; *Sternfeld et al., 2004*), produced a significant reduction of uIPSCs that was close to the maximal potency of the drug (~40%; *Dawson et al., 2006*; *Figure 1b*, *Table 1*). IPSC rise and decay times were not affected by α5IA application (data not shown).

Interestingly, MC-mediated uIPSC amplitudes were significantly larger and failure rate of synaptic responses evoked by the first action potential was significantly smaller in Ts compared to Eu mice. Moreover, the total amount of synaptic charge (Q) transferred during a train of 5 action potentials was near 5-fold larger in Ts than in Eu (*Figure 1c*, *Table 1*). MC-PN uIPSCs exhibited faster rise time in Ts as compared to Eu mice. Yet, uIPSC decay time-constants were similar in the two genotypes (*Figure 1—figure supplement 2*, *Table 1*). We then examined glutamatergic recruitment of MCs by PNs and found that it was stronger in Ts than Eu mice. Unitary excitatory postsynaptic currents (uEPSCs) in connected PN-MC pairs exhibited larger amplitude, lower failure rate and larger charge transfer in Ts than Eu mice (*Figure 1d*, *Table 2*).

Importantly, in both Ts and Eu mice, short-term dynamics of MC-PN GABAergic and PN-MC glutamatergic synaptic transmission were depressing and facilitating, respectively. However, short-term depression of MC-dependent dendritic inhibition onto PNs was more pronounced in Ts than Eu mice (*Figure 1—figure supplement 3*). Conversely, glutamatergic recruitment of MCs displayed the classical strong facilitating characteristics (*Silberberg and Markram, 2007*) with no differences in short-term uEPSC dynamics in the two genotypes (*Figure 1—figure supplement 3*). Regardless of whether unitary synaptic responses exhibited significant alterations of short-term plasticity, both

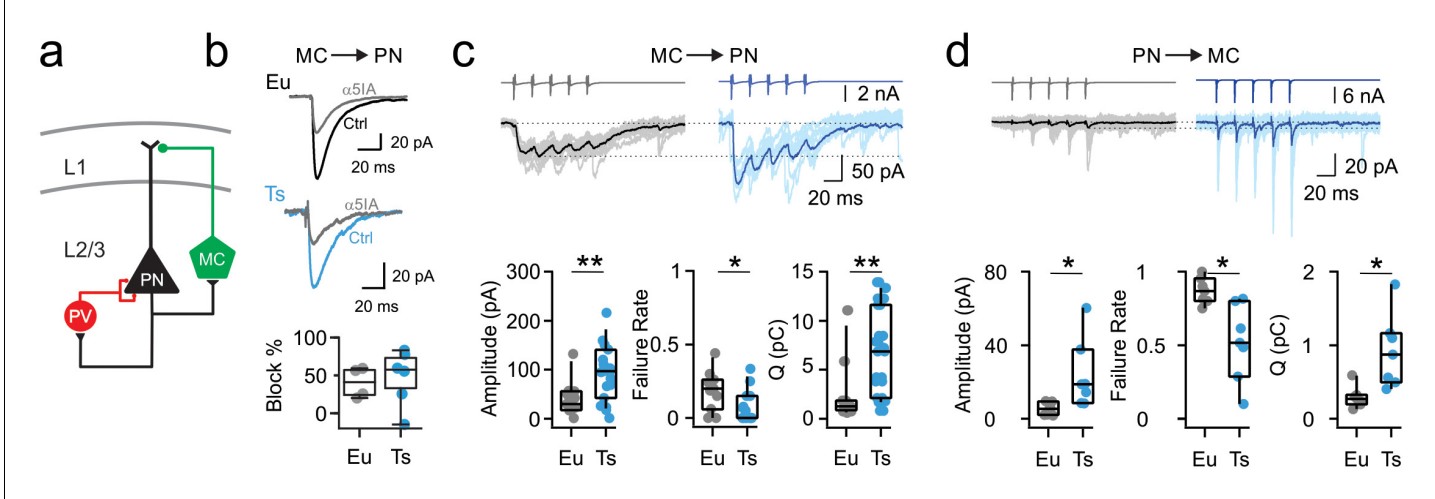

**Figure 1.** Synaptic enhancement of dendritic inhibition in DS. (a) Schematic of cortical inhibitory circuits involving PV interneurons, MCs and PNs. (b) Representative voltage-clamp current traces of uIPSCs recorded in MC-PN connected pair in a Eu (top) and a Ts (bottom) mouse, before (Ctrl) and after (α5IA) application of the α5-GABA$_A$R inverse agonist α5IA (100 nM). Shown are averages of 50 traces. Right, bottom: population data of uIPSC amplitude block by α5IA in Eu and Ts mice. (c) Top: Representative traces of uIPSCs (lower traces) elicited by a train of 5 presynaptic action currents (50 Hz) in the MC (upper traces). Eu: individual (gray) and average (black) traces; Ts: individual (light blue) and average (blue) traces. Bottom panels: population data of uIPSC amplitude (p=0.0094, Mann-Whitney U-test), failure rate of the postsynaptic response evoked by the first action potential of the train (p=0.0201, Mann-Whitney U-test) and total charge, transferred during the 5 APs train (Q, p=0.006, Mann-Whitney U-test; n = 11 and 15 pairs for Eu and Ts, respectively). (d) Same as in c, but for glutamatergic uEPSCs triggered by action currents in presynaptic PNs and recorded in postsynaptic MCs (Amplitude: p=0.0398, Mann-Whitney U-test; Failure Rate: p=0.0107, Mann-Whitney U-test; Q: p=0.0157, Students T-test; n = 7 for both, Eu and Ts).

The online version of this article includes the following source data and figure supplement(s) for figure 1:

**Source data 1.** Synaptic properties of the dendritic inhibitory loop.
**Figure supplement 1.** GFP positive interneurons in the prefrontal cortex of Ts::X98 mice are Martinotti cells.
**Figure supplement 2.** Abnormal IPSC kinetics in DS.
**Figure supplement 2—source data 1.** MC to PN synapse kinetics.
**Figure supplement 3.** Short term plasticity of the dendritic inhibitory loop in Ts.
**Figure supplement 3—source data 1.** Short term plasticity of the dendritic inhibitory loop.

MC-PN uIPSCs and PN-MC uEPSCs of Ts mice were characterized by a significant decrease of failure rate in all synaptic responses within the train. These results on short-term plasticity suggest that, in Ts mice, GABAergic and glutamatergic synapses involved in the PN-MC-PN dendritic inhibitory loop increased their efficacy using different pre- and postsynaptic strategies.

Overall, these results indicate that the dendritic inhibitory loop involving MCs and PNs is strengthened in Ts mice. Both output GABAergic synapses from MCs and their recruitment by local

**Table 1.** MC to PN synaptic efficiency evaluation.

| | | Dendritic Inhibition | | | | | | | | | |
|---|---|---|---|---|---|---|---|---|---|---|---|
| | | **Euploid** | | | | **Ts65Dn** | | | | | |
| | | median | Q1 | Q3 | n | median | Q1 | Q3 | n | test | p value |
| MC-PN IPSC | Amplitude (pA) | 29.89 | 20.24 | 52.98 | 11 | 97.09 | 54.49 | 135.86 | 15 | MW-test | 0.0095 |
| | Failure Rate | 0.20 | 0.11 | 0.25 | 11 | 0.00 | 0.00 | 0.13 | 15 | MW-test | 0.0201 |
| | Charge (pC) | 1.28 | 0.96 | 1.84 | 11 | 6.86 | 2.96 | 8.32 | 15 | MW-test | 0.00595 |
| | α5IA % block | 41 | 24.20 | 57.20 | 4 | 57.4 | 33.10 | 73.10 | 6 | MW-test | 0.2278 |
| | Rise Time (ms) | 1.20 | 1.00 | 1.80 | 18 | 1.70 | 1.50 | 2.10 | 18 | MW-test | 0.02448 |
| | Decay Time (ms) | 15.60 | 10.00 | 19.40 | 18 | 17.10 | 15.00 | 21.70 | 18 | MW-test | 0.1488 |

Median: quantile 50; Q1: quantile 25; Q3: quantile 75; n: number of cells; IPSC: Inhibitory postsynaptic current.

**Table 2.** PN to MC synaptic efficiency evaluation.

| | | Euploid | | | | Ts65Dn | | | | | |
|---|---|---|---|---|---|---|---|---|---|---|---|
| | | median | Q1 | Q3 | n | median | Q1 | Q3 | n | test | p value |
| PN-MC EPSC | Amplitude (pA) | 5.36 | 2.13 | 7.46 | 7 | 18.66 | 11.44 | 28.21 | 7 | T-test | 0.0106 |
| | Failure Rate | 0.87 | 0.81 | 0.93 | 7 | 0.52 | 0.39 | 0.71 | 7 | MW-test | 0.0126 |
| | Charge (pC) | 0.27 | 0.22 | 0.31 | 7 | 0.87 | 0.53 | 1.13 | 7 | MW-test | 0.0073 |

Median: quantile 50; Q1: quantile 25; Q3: quantile 75; n: number of cells; EPSC: Excitatory postsynaptic current.

glutamatergic synapses were stronger and more reliable in Ts mice, as compared to their euploid littermates.

## Excitability and morphology of MCs and PNs in Ts mice

Alteration of the MC-PN-MC synaptic loop can be associated to changes in intrinsic excitability and morphological features. We therefore tested whether passive properties, single action potentials and firing dynamics were altered in both PNs and MCs. In addition, we filled neurons with biocytin and we quantified their dendritic and axonal arborizations. Input-output spiking activity of both PNs and MCs was assessed by injecting increasing depolarizing 2 s-long currents. The firing frequency vs. injected current (*f-i*) curve was similar in both cell types in Eu and Ts mice (*Figure 2a,b*; *Tables 3–4*). Furthermore, single-action potential features, and passive properties were similar in both genotypes except for a small but significant increase in action potential threshold of Ts PNs (*Figure 2—figure supplements 1* and *2*; *Tables 5–6*). Importantly, the density of GFP-expressing MCs, and, in general, of SST-positive interneurons was similar in both genotypes (*Figure 2c*; *Figure 2—figure supplement 3*).

Increased MC-PN GABAergic transmission synaptic transmission in Ts mice can be attributed to axonal sprouting of MCs and/or increased dendritic branching of PNs. We performed a morphometric analysis of both cell types and found that the spatial distribution and total length of axons and dendrites of MCs were similar in both genotypes (*Figure 2d–f*). Likewise, both apical and basal dendrite arborizations of PNs were indistinguishable in Eu and Ts mice (*Figure 2g–i*).

These experiments and those illustrated in *Figure 1* indicate that the increased dendritic inhibitory loop involving MCs and PNs can be largely attributable to alteration of synaptic transmission between these two cell types.

## Excitability of PV cells, and not their perisomatic control of PNs, is strongly altered in Ts mice

Is the synaptic enhancement of the dendritic inhibitory loop involving MCs a specific alteration or a general feature of glutamatergic and GABAergic synapses in Ts mice? To address this question, we measured glutamatergic recruitment onto, and synaptic inhibition from, another prominent interneuron class, the PV basket cell. This interneuron class is characterized by its ability to fire high frequency, non-adapting trains of fast action potentials. These properties, along with perisomatic synaptic targeting of PNs, make the PV cell an efficient regulator of PN output. We thus crossed Ts65Dn with PValb-tdTomato mice, a line that expresses TdTomato specifically in PV-positive interneurons (*Kaiser et al., 2016*). We recorded uIPSCs and uEPSCs (*Figure 3a–d*) from pairs of synaptically connected PNs and PV cells. The amplitude, failure rate and charge transfer of uIPSC trains evoked by action potentials in presynaptic PV interneurons were similar in both genotypes (*Figure 3a,b*; *Table 7*). Likewise, the amplitudes, failure rates and charge transfer of uEPSCs elicited by PN firing were indistinguishable in Eu and Ts mice (*Figure 3c,d*; *Table 8*). Consistently, no difference was observed in short-term plasticity of both uIPSCs and uEPSCs in Ts and Eu mice (*Figure 3—figure supplement 1*).

Surprisingly, however, we found that intrinsic excitability of PV cells of Ts mice was strongly altered. These interneurons required one fourth less current (rheobase) necessary to induce firing action potentials. Accordingly, the gain of the f-I curves of Ts mice was dramatically reduced, as compared to their Eu littermates (*Figure 4a–c*; *Table 9*). Moreover, PV cells in Ts mice could not

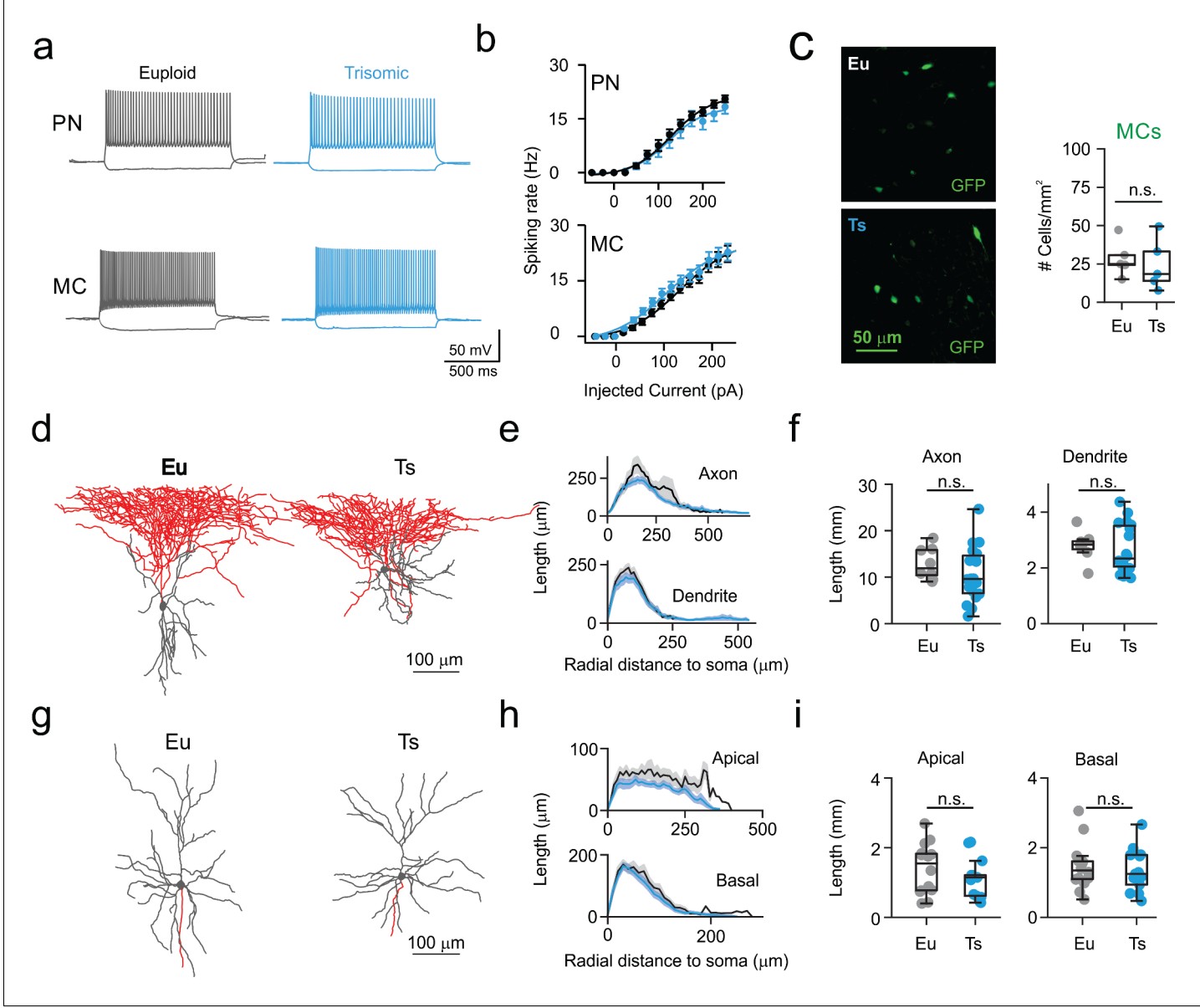

**Figure 2.** Normal excitability and morphology of PNs and MCs in Ts mice. (**a**) Representative current-clamp traces of membrane potential responses to injections of current steps of increasing amplitude applied to PNs (above) and MCs (below) from Eu (gray) and Ts (blue) mice. (**b**) Spiking frequency as function of injected current. Population data from PNs (left, genotype factor $F_{(1, 598)}$=3.444, p=0.064, two-way ANOVA; n = 25 and 23 cells for Eu and Ts, respectively) and MCs (right, genotype factor $F_{(1, 660)}$=0.004960, p=0.9439, two-way ANOVA; n = 26 and 20 cells for Eu and Ts, respectively). (**c**) Count of MC somas in the mPFC of Eu and Ts mice. Left: epifluorescence images of immuno-labeled GFP-expressing MCs in Ts::X98 coronal slices. Right: population data for both Eu and Ts. (**d**) Representative reconstruction of biocytin filled L2/3 MCs from Eu and Ts mice. Gray: somatodendritic region, red: axon. (**e**) Scholl analysis of MC axonal and dendritic length between concentric circles of increasing radial steps of 10 µm. (**f**) Population data of total axonal (left) and dendritic (right) length. (**e,f**) n = 18 and 8 neurons for Eu and Ts, respectively. (**g**) Representative reconstruction of biocytin filled PNs from Eu and Ts mice. Gray: somatodendritic region (including apical and basal dendrites), red: axon. (**h–i**) same as in e-f, but for apical and basal dendrites of PNs ($F_{(1, 25)}$=2.487, p=0.1273 2-way ANOVA for apical dendrites; $F_{(1, 25)}$=0.3521, p=0.5583 2-way ANOVA for basal dendrites). (**h,i**) n = 14 and 13 neurons for Eu and Ts, respectively. (**c,f,i**) Boxplots represent median, percentiles 25 and 75 and whiskers are percentiles 5 and 95. Points represent values from individual synapses (**b,c**), mice (**f**), neurons (**i,l**). *: p<0.05; **: p<0.01.

The online version of this article includes the following source data and figure supplement(s) for figure 2:

**Source data 1.** Excitability and morphology of PNs and MCs.
**Figure supplement 1.** Single action potential and passive properties are normal Ts mPFC PNs.
**Figure supplement 1—source data 1.** Singleaction potentialand passive properties of mPFC PNs.
**Figure supplement 2.** Active and passive properties are normal in mPFC MCs from Ts65Dn mice.

*Figure 2 continued on next page*

*Figure 2 continued*

**Figure supplement 2—source data 1.** Singleaction potentialand passive properties of mPFC MCs.
**Figure supplement 3.** Normal SST-positive interneurons density in the mPFC of Ts mice.

sustain high-frequency firing in response to 2 s-long depolarization, and the maximal spike rate was near half of that reached by PV cells in Eu mice (*Figure 4a–c*; *Table 9*). Notably, in Ts mice, action potential width was 1.7-fold wider and input resistance was 1.9-fold larger than that observed in Eu mice (*Figure 4d,e,*). Conversely, action potential threshold and amplitude were not affected (*Figure 4—figure supplement 1*; *Table 10*). Abnormal passive and active properties of PV cells of Ts mice were present throughout all ages under study. Importantly, in Eu mice, PV-cell active and passive properties reached values similar to those reported for this cell type (*Kaiser et al., 2016*; *Figure 4—figure supplement 2*). Similarly to SST cells, the density of PV-INs in mPFC was similar in the two genotypes (*Figure 4f*).

Altogether, these results indicate that, contrary to dendritic inhibition, the synaptic efficiency of the perisomatic feedback loop mediated by PV-INs was normal in Ts mice. Yet, PV-INs of Ts mice lost their characteristic electrophysiological fast-spiking signature, and their excitability was dramatically increased.

## Reduced spiking activity in vivo and increased tuning with network oscillations in Ts65Dn mPFC

Increased dendritic inhibition by the MC-PN loop, altered spiking activity of PV cells and their increased excitability will likely strongly influence the spiking properties and dynamics of mPFC PNs in vivo during spontaneous network activity. In order to assess the activity of the mPFC in vivo, we performed simultaneous local field potential (LFP) and loose-patch, juxtacellular recordings from layer 2/3 putative PNs to monitor their spiking dynamics related to overall network activity (*Figure 5a*). In vivo recordings exhibited typical oscillatory activity consisting of UP and DOWN states (*Ruiz-Mejias et al., 2011*). UP and DOWN states were similar in frequency and duration in Ts mice and their Eu littermates (*Figure 5—figure supplement 1*). UP states were enriched in γ-band activity (30–100 Hz) and exhibited increased probability of spiking activity (*Ruiz-Mejias et al., 2011*; *Figure 5a*). Juxtacellular recordings from individual mPFC putative PNs revealed a near 50% decrease in the overall spiking rate in Ts mice as compared to their Eu littermates (*Figure 5b*, *Table 11*). This difference was explained by a significant reduction of spiking rate during UP but not DOWN states (data not shown). Analysis of LFP power spectral density (PSD) did not show a significant difference in the two genotypes (*Figure 5c*). Interestingly, however, when we analyzed LFP waveform specifically during periods of neuronal spiking activity (spike-triggered LFP or stLFP), we found that, in both genotypes, average stLFPs exhibited marked voltage deflections, indicating that spike probability was not randomly distributed but locked to LFP oscillations (*Figure 5d*). The peak-to-peak amplitude of the stLFP was much larger in Ts than in Eu mice (*Figure 5d,e*, *Table 11*), and, remarkably, the spectral power of the stLFP was largely increased in Ts mice, selectively in the β-γ-frequency band (*Figure 5f*). In order to quantitatively assess whether PN spikes were differently locked to the phase of network oscillations, we measured the pairwise phase consistency (PPC), which is an unbiased parameter to determine the degree of tuning of single-neuron firing to network rhythmic activity of specific frequencies (*Perrenoud et al., 2016*; *Veit et al., 2017*; *Vinck et al.,*

**Table 3.** PNs excitability.

| | | Euploid | | | | Ts65Dn | | | | | |
|---|---|---|---|---|---|---|---|---|---|---|---|
| | | median | Q1 | Q3 | n | median | Q1 | Q3 | n | test | p value |
| Excitability | InjCurr50 (pA) | 21.6 | 19.5 | 24.4 | 12 | 18.9 | 15.9 | 24.6 | 11 | T-test | 0.6081 |
| | Max Spiking Rate (Hz) | 112.4 | 91.9 | 155.0 | 12 | 124.5 | 101.0 | 178.2 | 11 | T-test | 0.4500 |

Median: quantile 50; Q1: quantile 25; Q3: quantile 75; n: number of cells; InjCurr50: amount of current injected to reach 50% of the maximal spiking rate.

**Table 4.** MCs excitability.

| | | Euploid | | | | Ts65Dn | | | | | |
|---|---|---|---|---|---|---|---|---|---|---|---|
| | | median | Q1 | Q3 | n | median | Q1 | Q3 | n | test | p value |
| Excitability | InjCurr50 (pA) | 40.9 | 26.9 | 50.4 | 25 | 48.9 | 25.4 | 70.2 | 20 | MW-test | 0.6073 |
| | Max Spiking Rate (Hz) | 101.8 | 68.3 | 124.0 | 25 | 86.5 | 63.2 | 109.7 | 20 | T-test | 0.4436 |

Median: quantile 50; Q1: quantile 25; Q3: quantile 75; n: number of cells; InjCurr50: amount of current injected to reach 50% of the maximal spiking rate.

*2010*). Ts mice exhibited significantly higher PPC values than Eu littermates for frequency bands ranging between 10 and 60 Hz (*Figure 4g*, *Table 12*), thus revealing stronger phase locking selectively with β- and low γ-oscillations.

Altogether, these results indicate that mPFC PNs in Ts mice fire less than their Eu littermates, but their spontaneous spiking activity is more strongly tuned to β- and γ-frequency bands. Reduced firing rate and increased phase locking with fast oscillations are both consistent with increased activity of inhibitory interneurons (*Cardin et al., 2009*; *Chen et al., 2017*; *Sohal et al., 2009*; *Veit et al., 2017*).

## Discussion

In this study, we analyzed synaptic and intrinsic properties of two major inhibitory circuits of the prefrontal cortex in a relevant mouse model of DS. We found specific alterations of distinct GABAergic circuits. In particular, we demonstrate that the dendritic inhibitory synaptic loop involving MCs and PNs was strongly potentiated in Ts, as compared to euploid control mice. In contrast, the perisomatic synaptic inhibitory control of PNs by PV cells was not affected in Ts mice. Strikingly, however, the excitability of PV cells was profoundly altered. These GABAergic circuit-specific alterations correlate with reduced PN spiking, and enhanced coupling with network β-γ-activity in Ts mice in vivo.

We analyzed MCs, which are a subset of SST interneurons, specialized in inhibiting the distal portion of apical dendrites of PNs, by ascending their axons in L1 (*Kawaguchi and Kubota, 1998*), where PN dendrites integrate top-down input originating from distal brain areas. In the hippocampus, dendritic inhibition strongly regulates PN dendrite electrogenesis and supra-linearity (*Lovett-Barron et al., 2012*), likely modulating the emergence of burst firing (*Royer et al., 2012*). Interestingly, α5–mediated dendritic inhibition in the hippocampus, known to strongly control dendritic integration and action potential firing (*Schulz et al., 2018*), is also enhanced in Ts65Dn mice and strongly control NMDAR activation, nonlinear dendritic integration, and AP firing (*Schulz et al., 2019*). In the neocortex, integration of top-down information in L1 and consequent dendrite-dependent generation of burst activity was hypothesized to underlie the encoding of context-rich, salient information (*Larkum, 2013*). Our results indicate a specific synaptic strengthening of the dendritic inhibitory loop involving MCs in Ts mice, suggesting a major impact on PN dendritic integration and electrogenesis.

Enhanced dendritic inhibition by MCs in DS could underlie the deficits of long-term plasticity of glutamatergic synapses similar to those observed in the hippocampus of Ts mice. Indeed, these LTP

**Table 5.** PNs passive and action potential (AP) properties.

| | | Euploid | | | | Ts65Dn | | | | | |
|---|---|---|---|---|---|---|---|---|---|---|---|
| | | median | Q1 | Q3 | n | median | Q1 | Q3 | n | test | p value |
| Passive properties | Vrest (mV) | -76 | -79 | -72 | 25 | -73 | -78 | -69 | 22 | T-test | 0.2473 |
| | Ri (MΩ) | 255 | 139 | 378 | 25 | 178 | 86 | 316 | 22 | T-test | 0.7185 |
| | Tau memb (ms) | 36.5 | 28.1 | 47.9 | 25 | 39.7 | 26.8 | 46.9 | 22 | T-test | 0.5790 |
| AP properties | Threshold (mV) | -40.9 | -45.6 | -39.2 | 25 | -39.6 | -40.8 | -38.5 | 22 | MW-test | 0.0453 |
| | Amplitude (mV) | 83.6 | 75.1 | 85.8 | 25 | 82.5 | 69.2 | 85.5 | 22 | MW-test | 0.3749 |
| | Width (ms) | 1.2 | 1.0 | 1.6 | 25 | 1.2 | 1.0 | 1.6 | 22 | MW-test | 0.7413 |

Median: quantile 50; Q1: quantile 25; Q3: quantile 75; n: number of cells; Vrest: Resting membrane potential; Ri : input resistance.

**Table 6.** MCs passive and action potential (AP) properties.

| | | Euploid | | | | Ts65Dn | | | | | |
|---|---|---|---|---|---|---|---|---|---|---|---|
| | | median | Q1 | Q3 | n | median | Q1 | Q3 | n | test | p value |
| Passive properties | Vrest (mV) | -66 | -70 | -63 | 26 | -64 | -67 | -62 | 20 | T-test | 0.0566 |
| | Ri (MΩ) | 238 | 183 | 382 | 26 | 207 | 176 | 282 | 20 | MW-test | 0.5721 |
| | Tau memb (ms) | 27.0 | 22.4 | 41.4 | 25 | 33.0 | 28.7 | 51.7 | 20 | MW-test | 0.1407 |
| AP properties | Threshold (mV) | 79.0 | 63.2 | 85.1 | 26 | 83.4 | 77.3 | 88.5 | 20 | MW-test | 0.0988 |
| | Amplitude (mV) | -45.8 | -51.5 | -42.2 | 26 | -42.9 | -44.6 | -42.0 | 20 | MW-test | 0.0727 |
| | Width (ms) | 1.2 | 0.9 | 1.6 | 26 | 1.2 | 0.9 | 1.5 | 20 | MW-test | 0.8680 |

Median: quantile 50; Q1: quantile 25; Q3: quantile 75; n: number of cells; Vrest: Resting membrane potential ; Ri : input resistance.

deficits can be recovered by treatment of allosteric modulators of α5-GABA$_A$Rs (*Duchon et al., 2020*; *Martínez-Cué et al., 2013*; *Schulz et al., 2019*). Likewise, the enhanced MC-PN inhibitory loop in Ts mice shown here can provide a mechanism for the rescue of cognitive deficits in Ts mice, operated by selective pharmacology of α5-GABA$_A$Rs (*Braudeau et al., 2011*; *Duchon et al., 2020*; *Martínez-Cué et al., 2013*). This subunit of GABA$_A$Rs, whose synaptic vs. extrasynaptic expression is still debated (*Ali and Thomson, 2008*; *Botta et al., 2015*; *Glykys and Mody, 2010*; *Hannan et al., 2020*; *Hausrat et al., 2015*; *Schulz et al., 2018*; *Serwanski et al., 2006*), mediate dendritic synaptic inhibition from neocortical MCs (*Ali and Thomson, 2008*) and their hippocampal counterparts (the oriens-lacunosum moleculare or O-LM interneurons, *Schulz et al., 2018*). Here, we confirmed that

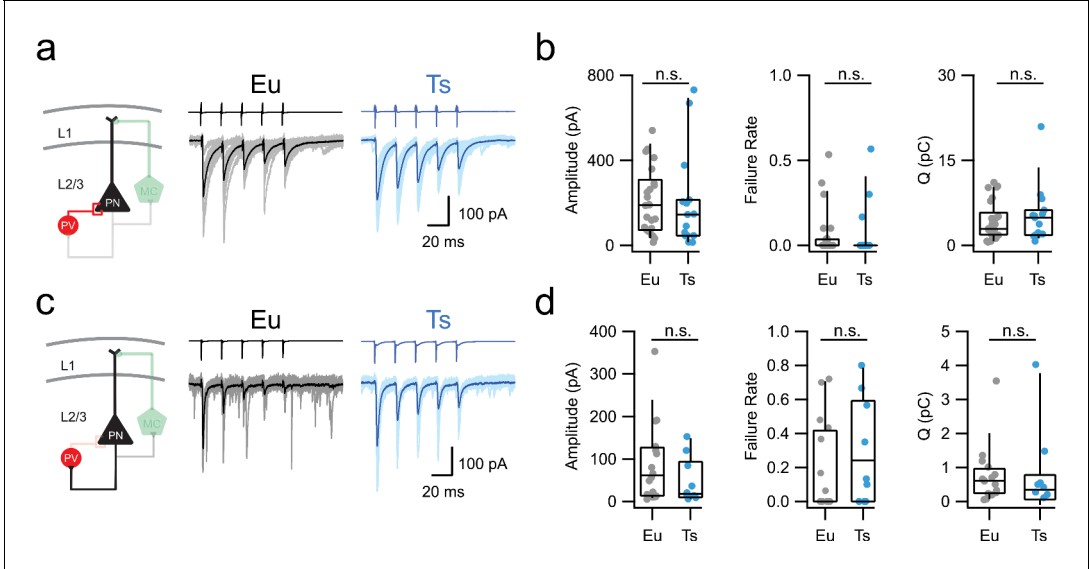

**Figure 3.** Perisomatic inhibition by PV-INs is normal in Ts mice. (**a**) Left: scheme of the PV-PN perisomatic inhibitory circuit assessed using dual whole-cell patch-clamp recordings. Right: representative voltage-clamp traces corresponding to uIPSCs evoked upon application of 5 action currents (50 Hz) to the presynaptic PV-IN. Eu: individual (gray) and average (black); Ts: individual (light blue) and average (blue) traces are superimposed. (**b**) Population data of uIPSC amplitude (p=0.4901, Mann-Whitney U-test), failure rate of the postsynaptic response evoked by the first action potential of the train (p=0.7185, Mann-Whitney U-test) and total charge transferred during the 5 APs train (Q, p=0.579, Mann-Whitney U-test, n = 26 and 15 pairs for Eu and Ts, respectively). (**c–d**) Same as in a-b, but for glutamatergic uEPSCs triggered in the presynaptic PN and recorded in the postsynaptic PV IN in both genotypes (Amplitude: p=0.233, Mann-Whitney U-test; Failure Rate: p=0.214, Mann-Whitney U-test; Q: p=0. 3711, Mann-Whitney U-test; n = 15 and 9 pairs for Eu and Ts, respectively).

The online version of this article includes the following source data and figure supplement(s) for figure 3:

**Source data 1.** Synaptic properties of the perisomatic inhibitory loop.

**Figure supplement 1.** Short term plasticity of the perisomatic inhibitory loop in Ts.

**Figure supplement 1—source data 1.** Short term plasticity of the perisomatic inhibitory loop in Ts.

**Table 7.** Direct perisomatic inhibition.
PV to PN synaptic efficiency evaluation.

| | | Perisomatic inhibition | | | | | | | | |
|---|---|---|---|---|---|---|---|---|---|---|
| | | Euploid | | | | Ts65Dn | | | | | |
| | | median | Q1 | Q3 | n | median | Q1 | Q3 | n | test | p value |
| PV-PN IPSC | Amplitude (pA) | 188.43 | 74.87 | 289.03 | 26 | 144.52 | 46.10 | 209.78 | 15 | MW-test | 0.49 |
| | Failure Rate | 0.00 | 0.00 | 0.03 | 26 | 0.00 | 0.00 | 0.00 | 15 | MW-test | 0.7185 |
| | Charge (pC) | 2.88 | 1.92 | 5.02 | 26 | 4.89 | 1.89 | 5.85 | 15 | MW-test | 0.579 |

PV to PN synaptic efficiency evaluation. Median: quantile 50; Q1: quantile 25; Q3: quantile 75; n: number of cells; IPSC: Inhibitory postsynaptic current.

α5-containing GABA$_A$Rs are majorly responsible for MC-PN dendritic synaptic inhibition, due to the overall block of α5IA, which was close to the max potency of this drug (*Dawson et al., 2006*).

The enhancement of the dendritic inhibitory loop involving MCs and PNs in Ts mice could not be attributed to alterations of axonal and/or dendritic arborizations in the mutant mice. On the other hand, it could be due to a combination of pre- and postsynaptic mechanisms, including alterations of release probability, number of release sites or quantal size at either GABAergic or glutamatergic synapses involved in this circuit. The strong increase of synaptic charge at MC-PN GABAergic synapses could be attributed to alterations of uIPSC waveform in Ts mice. Although the small change of uIPSC rise time can in principle account for, at least in part, an increase in synaptic charge, it is unlikely to account for the 5-fold increase of synaptic charge, with no changes in uIPSC decay.

The robust increase of synaptic charge in Ts mice is likely due to one or a combination of different non-linearities that dynamically emerge during a spike train. These include the recruitment of peri- or extrasynaptic GABA$_A$Rs, possible alterations of release probability or quantal size during trains and differences in postsynaptic dendritic filtering in the two genotypes. These complex interplays between these factors occur at distal dendrites; therefore, they are difficult to tease out from somatic recordings as they are characterized by poor dendritic voltage control due to space-clamping constraints. The enhanced short-term depression of MC-PN uIPSCs that we report here suggests that release probability at dendritic GABAergic synapses from MCs is increased in Ts mice. This is consistent with increased short-term depression of inhibition reported in the dentate gyrus (*Kleschevnikov et al., 2012*), but not with unaltered short-term plasticity shown in CA1 (*Mitra et al., 2012*). These discrepancies could be ascribed to differences between brain regions. Moreover, they could also be a consequence of non-specific recruitment of presynaptic axons by global extracellular stimulation, in contrast to the isolation of unitary synaptic responses by dual intracellular recordings as reported here. The lack of short-term plasticity alterations at PN-MC glutamatergic synapses suggests that increased excitatory recruitment of MCs is due to alterations at postsynaptic sites. Future studies will be necessary to pinpoint the exact biophysical and anatomical alterations underlying the prominent increase of dendritic inhibition operated by MCs in DS.

Potentiation of glutamatergic synapses in Ts mice seems to be specific for PN-MC connections, as PN-PV synapses were similar in both genotypes, suggesting that presynaptic terminals of local

**Table 8.** Recruitment of PV-INs by PNs.
PN to PV synaptic efficiency evaluation.

| | | Perisomatic inhibition | | | | | | | | |
|---|---|---|---|---|---|---|---|---|---|---|
| | | Euploid | | | | Ts65Dn | | | | | |
| | | median | Q1 | Q3 | n | median | Q1 | Q3 | n | test | p value |
| PN-PVEPSC | Amplitude (pA) | 65.66 | 19.28 | 124.50 | 15 | 20.41 | 14.10 | 84.25 | 9 | MW-test | 0.23304 |
| | Failure Rate | 0.00 | 0.00 | 0.27 | 15 | 0.13 | 0.00 | 0.57 | 9 | MW-test | 0.21402 |
| | Charge (pC) | 0.62 | 0.29 | 0.90 | 15 | 0.42 | 0.20 | 0.55 | 9 | MW-test | 0.3711 |

Median: quantile 50; Q1: quantile 25; Q3: quantile 75; n: number of cells; EPSC: Excitatory postsynaptic current.

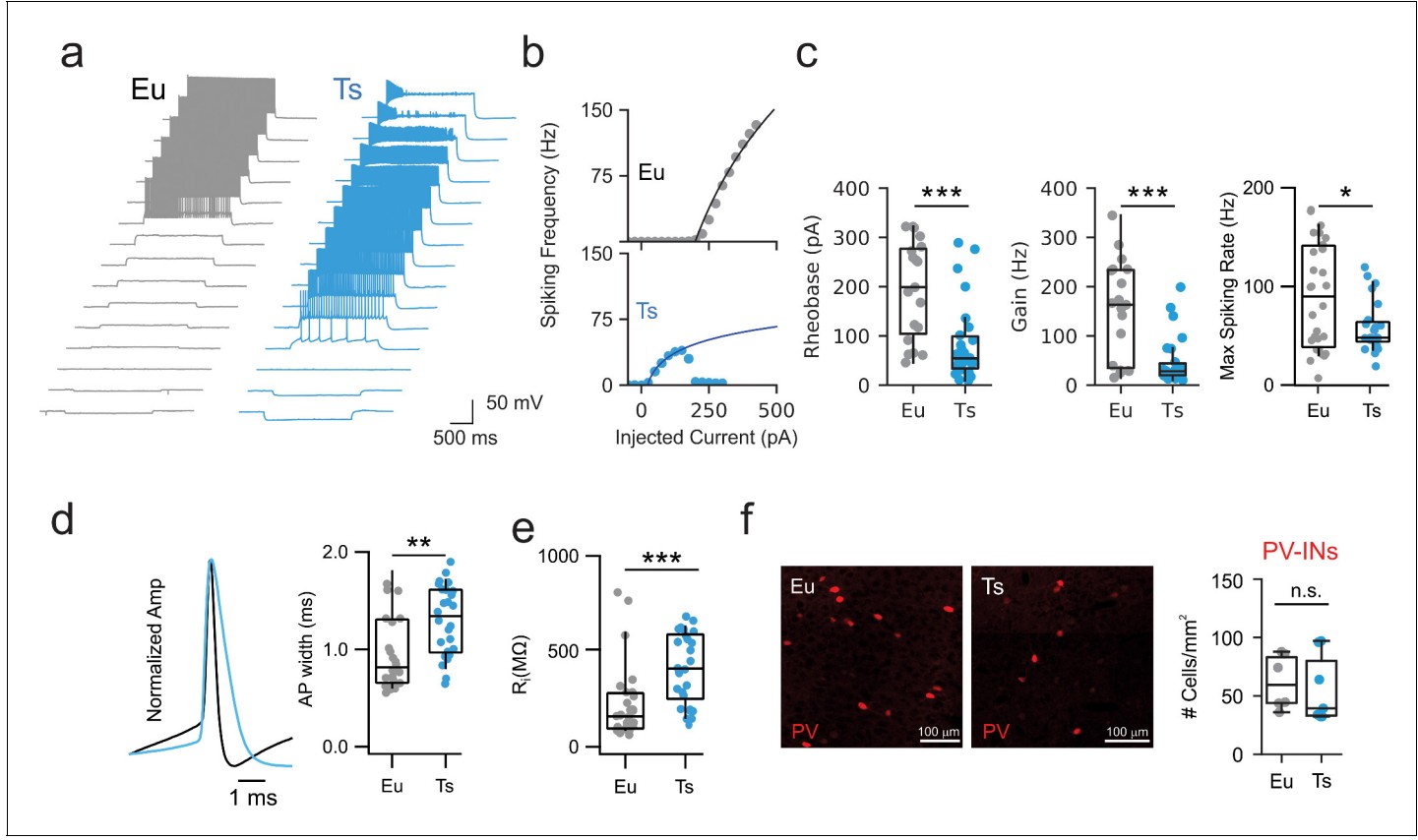

**Figure 4.** Altered excitability of PV cells in Ts mice. (**a**) Representative voltage traces in response to current steps of increased amplitudes applied to to PV-INs from Eu (gray) and Ts (blue). (**b**) Representative F-I curves recorded from Eu (top) and Ts (bottom) individual interneurons. Continuous lines represent logarithmic fit used to estimate Rheobase and Gain in each recorded cell. (**c**) Left: Rheobase population data for Eu (194.9 ± 22.4 pA) and Ts mice (85.0 ± 15.3 pA; p=0.00038, Mann-Whitney U-test). Middle: Gain population data for Eu (150.8 ± 22.6 Hz) and Ts mice (47.2 ± 9.4 Hz; p=0.00035, Mann-Whitney U-test). Right: Maximal spiking rate reached upon current injection (p=0.04, Mann-Whitney U-test, n = 28 and 26 cells for Eu and Ts, respectively). (**d**) Left: representative action-potential traces, scaled to the peak, from Eu (black) and Ts (blue) mice. Right: population data of AP width in the two genotypes. (**e**) Population data of input resistance measured in Eu and Ts PV-INs. (**f**) Quantification of PV-INs somas in the mPFC of Eu and Ts mice. Left: Epifluorescence images of immunolabeled PV cells in coronal slices from Ts::PV mice. Right: density of PV cells Eu and Ts (n = 6 and 7 mice for Eu and Ts, respectively; p=0.2602, Mann-Whitney test). *: p<0.05; **: p<0.01.

The online version of this article includes the following source data and figure supplement(s) for figure 4:

**Source data 1.** Excitability of PV cells.
**Figure supplement 1.** Specific alteration of mPFC PV-INs electrical properties in Ts65Dn mice.
**Figure supplement 1—source data 1.** Singleaction potentialand passive properties of mPFC PV cells.
**Figure supplement 2.** PV cells passive and active properties as function of Age.

glutamatergic synapses can undergo target-specific modulation of their strength. Intriguingly, it has been recently shown that an increase of the excitatory drive of hippocampal interneurons (due to triplication of GluR5 kainate receptor expression) could explain excess of inhibition received by

**Table 9.** PV-INs excitability.

| | | Euploid | | | | Ts65Dn | | | | | |
| --- | --- | --- | --- | --- | --- | --- | --- | --- | --- | --- | --- |
| | | median | Q1 | Q3 | n | median | Q1 | Q3 | n | test | p value |
| Excitability | Rheobase (pA) | 198.7 | 104.1 | 276.7 | 19 | 54.4 | 34.6 | 98.7 | 26 | MW | 0.0004 |
| | Gain (Hz) | 163.2 | 34.6 | 233.8 | 19 | 28.2 | 20.2 | 44.2 | 26 | MW | 0.0004 |
| | Max Spiking Rate (Hz) | 99.5 | 45.2 | 142.3 | 19 | 47.9 | 45.3 | 63.3 | 26 | MW | 0.0001 |

Median: quantile 50; Q1: quantile 25; Q3: quantile 75; n: number of cells; InjCurr50: amount of current injected to reach 50% of the maximal spiking rate.

**Table 10.** PV-INs passive and action potential (AP) properties.

| | | Euploid | | | | Ts65Dn | | | | | |
|---|---|---|---|---|---|---|---|---|---|---|---|
| | | median | Q1 | Q3 | n | median | Q1 | Q3 | n | test | p value |
| Passive properties | Vrest (mV) | -71 | -75 | -69 | 28 | -70 | -71 | -66 | 26 | T-test | 0.1808 |
| | Ri (MΩ) | 154 | 93 | 271 | 28 | 406 | 264 | 581 | 26 | MW-test | 0.0001 |
| | Tau memb (ms) | 12.0 | 6.5 | 20.6 | 28 | 25.8 | 20.2 | 37.3 | 25 | MW-test | 0.0003 |
| AP properties | Threshold (mV) | -43.4 | -47.0 | -39.5 | 26 | -44.0 | -46.0 | -41.3 | 26 | T-test | 0.7891 |
| | Amplitude (mV) | 65.2 | 54.6 | 70.3 | 25 | 69.0 | 62.8 | 73.4 | 25 | T-test | 0.1019 |
| | Width (ms) | 0.8 | 0.7 | 1.3 | 26 | 1.3 | 1.0 | 1.6 | 26 | MW-test | 0.0012 |

Median: quantile 50; Q1: quantile 25; Q3: quantile 75; n: number of cells; Vrest: Resting membrane potential; Ri : input resistance.

pyramidal neurons in the Ts2Cje Down syndrome mouse model (*Valbuena et al., 2019*). Therefore, a similar gene overdose of kainate receptors can boost the recruitment of specific interneurons in the PFC of Ts mice.

The strong increase of membrane resistance of PV cells in Ts mice underlies the augmented intrinsic excitability and can produce early ectopic bouts of activity, thus contributing to network over-inhibition. Enhanced membrane resistance could result from alterations in the expression of TWIK1 and TASK1 leak channels. Indeed, these channels underlie the developmental decrease of membrane resistance in PV cells (*Okaty et al., 2009*). In contrast, the inability of PV cells of Ts mice to sustain high frequency firing could prevent these interneurons from generating high-frequency bursts of action potentials and therefore have a detrimental effect on the temporal coding of these interneurons.

Interestingly, however, despite the dramatic alterations of intrinsic excitability in Ts PV cells, their output synaptic perisomatic inhibition was similar in both genotypes. This, despite the widening of single PV-cell action potentials, which could, in principle change the presynaptic $Ca^{2+}$ dynamics and thus alter release probability. Since action potentials are recorded in the soma, the lack of effect at PV-cell synapses could be due to soma-specific alterations. Alternatively, since neocortical PV-PN GABAergic synapses exhibit high release probability (*Deleuze et al., 2019*; *Kawaguchi and Kubota, 1998*), alterations of spike width might not be enough to produce additional increases. Increase of action potential width in Ts mice could be due to changes in the expression Kv3.1b potassium channels, known to underlie the fast repolarization of action potentials and fast-spiking behavior of PV cells (*Erisir et al., 1999*). Future experiments will be necessary to reveal the exact molecular mechanism underlying the altered firing properties of PV cells, which, in Ts mice, have lost their characteristic fast-spiking signature.

The aberrant active and passive properties of PV cells in Ts mice could be due to a delayed development of these interneurons. Indeed, during development, neocortical PV cells display marked accelerations of single action potentials and increased firing frequency, accompanied by decreased input resistance (*Okaty et al., 2009*). Although we cannot rule out the possibility of a delayed development of PV cells in Ts mice, these interneurons showed abnormal passive and active properties within the entire age range studied here. It will be interesting to determine whether these profound alterations of PV-cell firing are present with the same incidence and magnitude along the entire life span of Ts mice. Although previous reports have shown higher number of inhibitory interneurons in the hippocampus (*Hernández-González et al., 2015*) and somatosensory cortex (*Aziz et al., 2018*; *Chakrabarti et al., 2010*) we failed to detect significant differences in the density of both SST- and PV-positive interneurons in the mPFC. This could be due to differences between brain regions. A systematic comparative analysis will be required to better understand the consequences of DS neurodevelopmental alterations of the cellular composition in different brain regions.

Both potentiation of the dendritic inhibitory loop and increased PV-cell excitability are consistent with the alterations of PN spiking activity that we recorded in vivo. Indeed, reduced spike rates and increased tuning in the β-γ-frequency range are both consistent with increased activity of inhibitory neurons (*Atallah et al., 2012*; *Cardin et al., 2009*; *Chen et al., 2017*; *Sohal et al., 2009*; *Veit et al.,*

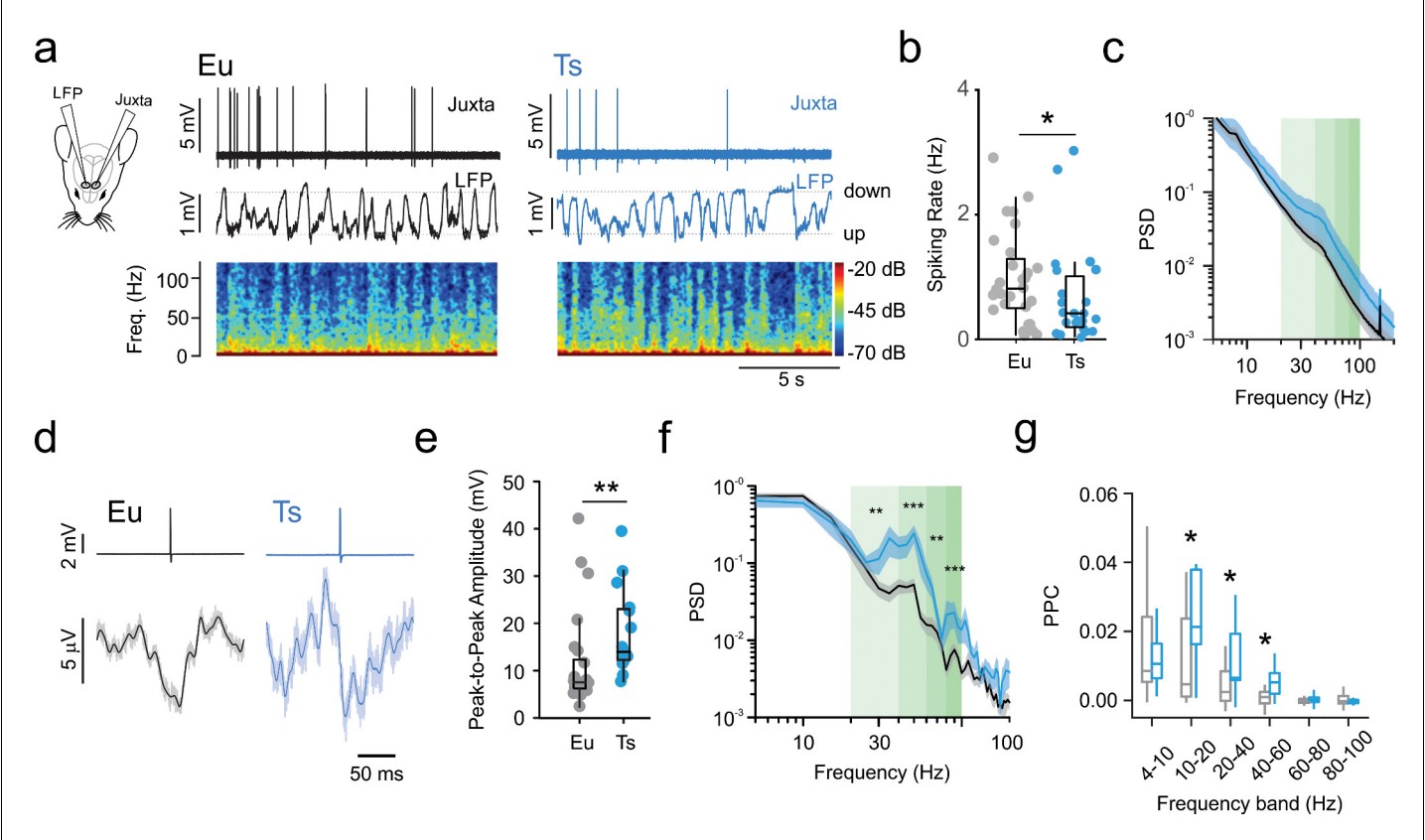

**Figure 5.** Reduced spiking activity in vivo and increased tuning with network oscillations in anesthetized Ts65Dn mPFC. (**a**) Left: scheme depicting simultaneous local field potential (LFP) and juxtacellular recordings in layer 2/3 of the mPFC. Right: representative juxtacellular (top trace), LFP (middle trace) and spectrogram (bottom) recorded in Eu (black traces) and Ts (blue traces) mice. (**b**) Average spiking rate from individual cells. Population data (n = 28 cells, 5 mice and 23 cells, six mice for Eu and Ts, respectively; p=0.03, Mann-Whitney U-test). (**c**) Normalized Power Spectral Density (PSD) from Eu and Ts mice. Shaded green areas correspond to β-γ-frequency ranges. (**d**) Representative portions of averaged LFP around aligned spike (spike-triggered LFP or stLFP). Top: Average traces of aligned spikes from a putative layer 2/3 PN recorded in a Eu (black) and Ts (blue) mouse. Bottom: average of the corresponding stLFPs. Light line: raw averaged trace, thick dark line: low pass filtered trace (cutoff: 100 Hz). (**e**) Average plots of stLFP peak-to-peak amplitude in Eu and Ts mice. (**f**) Normalized Power spectral density of stLFPs. Shaded green areas correspond to β-γ-frequency ranges. (**g**) Pairwise Phase Consistency (PPC) calculated for specific frequency bands (4–10 Hz: p=0.427, 10–20 Hz: p=0.02145, 20–40 Hz: p=0.0418, 40–60 Hz: p=0.01638, 60–80 Hz: p=0.2308, 80–100 Hz: p=0.3161, Mann-Whitney U-test). e-g,: n = 23 cells, 5 mice and 11 cells, six mice for Eu and Ts, respectively. Boxplots represent median, percentiles 25 and 75 and whiskers are percentiles 5 and 95. *: p<0.05; **: p<0.01.
The online version of this article includes the following source data and figure supplement(s) for figure 5:

**Source data 1.** In vivo spiking activity and phase locking.
**Figure supplement 1.** Characterization of slow wave oscillatory activity of LFP in anesthetized Eu and Ts mice.

*2017*). The overall LFP power spectra observed in mPFC is similar in both Ts and Eu mice, consistently with a recent report (*Chang et al., 2020*). However, close examination of stLFP revealed a strong increase in the power of β- and low γ-frequency bands during periods of spiking activity. This likely reflects the consequences of alterations at the level of local microcircuits.

We cannot directly link the inhibitory circuit-specific alterations that we detected in slices with the increased synchronization of β-γ-activity that we measured in vivo. However, a large body of literature indicates that oscillations in this frequency range strongly depends on the activity of PV cells (*Buzsáki and Wang, 2012*; *Sohal et al., 2009*; *Cardin et al., 2009*). More recently, also SST interneurons were shown to control PN phase coupling with low frequency (30 Hz) neocortical γ-oscillations (*Chen et al., 2017*; *Veit et al., 2017*). It is therefore tempting to speculate that increased dendritic inhibition from SST-expressing MCs modulates phase coupling of PNs with β- and low γ-

**Table 11.** LFP and single cell spiking recorded in vivo.

| | | Euploid | | | | | Ts65Dn | | | | | | |
|---|---|---|---|---|---|---|---|---|---|---|---|---|---|
| | | median | Q1 | Q3 | N cells | N mice | median | Q1 | Q3 | N cells | N mice | test | p value |
| | Spiking Rate (Hz) | 0.81 | 0.56 | 1.24 | 28 | 6 | 0.42 | 0.22 | 0.92 | 23 | 5 | MW-test | 0.0348 |
| stLFP | peak-to-peak Amp (µV) | 7.3 | 6.0 | 8.6 | 22 | 6 | 13.3 | 12.3 | 17.1 | 11 | 5 | MW-test | 0.0011 |

Median: quantile 50; Q1: quantile 25; Q3: quantile 75; n: number of cells.

oscillations. On the other hand, the increased phase coupling of PN spikes with high-frequency γ-activity could result from the augmented excitability of PV cells. The differential phase coupling of these two interneuron types at distinct frequencies is consistent with the peculiar fast and slower recruitment and biophysical properties of PV interneurons and MCs, respectively. Future experiments involving chemogenetic alterations of PV-interneuron excitability and/or pharmacological manipulations of MC-PN synapses in Ts mice will help decipher the role played by each inhibitory cell subtype in controlling the temporal dynamics of PN firing during different rhythmic cortical network activities. Alternatively, our results could be interpreted as differences in long-range functional connectivity in the two genotypes, possibly due to alterations in myelination and conduction velocity in Ts vs. Eu mice (*Olmos-Serrano et al., 2016a*). Nevertheless, a reduction of axonal conduction velocity would produce a temporal shift in spike-to-phase association, rather than increased phase locking.

The increase in β-γ-band power and enhanced neural synchronization in these frequency ranges in Ts mice is consistent with recent evidence indicating augmented hippocampal-PFC synchronization and LFP γ-band power during natural non-REM sleep in Ts65Dn mice (*Alemany-González et al., 2020*). This suggests that the alterations of γ-oscillations observed here could play a role in the pathophysiology of sleep disruptions reported in DS children (*Fernandez et al., 2017*) and adults (*Giménez et al., 2018*).

In sum, here we report direct evidence for over-inhibition of mPFC circuits in a mouse model of DS. However, over-inhibition was not due to a generic increase of GABAergic signaling, but emerged from highly specific synaptic and intrinsic alterations of dendritic and somatic inhibitory circuits, respectively. Future experiments are necessary to reveal whether other inhibitory neuron types are also affected in DS. Likewise, it will be fundamental to assess whether specific dysfunctions of individual GABAergic circuits underlie different aspects of cognitive deficits (e.g. impaired memory and flexibility, autistic traits), which affect individuals with DS.

**Table 12.** Pairwise Phase Consistency (PPC) descriptive statistics and hypothesis tests between genotypes for each frequency band analyzed.

| | | Euploid | | | | | Ts65Dn | | | | | | |
|---|---|---|---|---|---|---|---|---|---|---|---|---|---|
| | Freq. band | median | Q1 | Q3 | N cells | N mice | median | Q1 | Q3 | N cells | N mice | test | p value |
| PPC | 4-10 Hz | 0.0085 | 0.0053 | 0.0242 | 22 | 6 | 0.0106 | 0.0064 | 0.0164 | 11 | 5 | MW-test | 0.4270 |
| | 10-20 Hz | 0.0046 | 0.0011 | 0.0237 | 22 | 6 | 0.0213 | 0.0163 | 0.0379 | 11 | 5 | MW-test | 0.0215 |
| | 20-40 Hz | 0.0024 | -0.0002 | 0.0085 | 22 | 6 | 0.0066 | 0.0059 | 0.0193 | 11 | 5 | MW-test | 0.0418 |
| | 40-60 Hz | 0.0009 | -0.0009 | 0.0025 | 22 | 6 | 0.0052 | 0.0019 | 0.0079 | 11 | 5 | MW-test | 0.0164 |
| | 60-80 Hz | -0.0003 | -0.0007 | 0.0004 | 22 | 6 | 0.0001 | -0.0005 | 0.0009 | 11 | 5 | MW-test | 0.2308 |
| | 80-100 Hz | -0.0002 | -0.0009 | 0.0013 | 22 | 6 | -0.0004 | -0.0008 | 0.0002 | 11 | 5 | MW-test | 0.3161 |

Median: quantile 50; Q1: quantile 25; Q3: quantile 75; n: number of cells. We considered only cells from which we recorded at least 250 action potentials in order to obtain reliable PPC values.

# Materials and methods

**Key resources table**

| Reagent type (species) or resource | Designation | Source or reference | Identifiers | Additional information |
|---|---|---|---|---|
| Genetic reagent (*M. musculus*) | C57BL/6-Tg(Pvalb-tdTomato)15Gfng/J (PValb-Tomato) | Jackson Laboratory | Stock #: 27395 RRID:MGI:5629295 | |
| Genetic reagent (*M. musculus*) | Tg(Gad1-EGFP)98Agmo/J, GAD67-GFP (X98) | Jackson Laboratory | Stock #: 6340 RRID:MGI:3715263 | |
| Genetic reagent (*M. musculus*) | B6EiC3Sn.BLiA-Ts(1716)65Dn/DnJ (Ts65Dn) | Jackson Laboratory | Stock #: 005252 RRID:MGI:2178111 | |

Experimental procedures followed national and European (2010/63/EU) guidelines, and have been approved by the authors' institutional review boards and national authorities. All efforts were made to minimize suffering and reduce the number of animals. Experiments were performed on >4 week old and 18- to 25-day-old mice for in vivo and ex vivo recordings, respectively. We used *B6EiC3Sn a/A-Ts(17 $^{16}$)65Dn/J* (also known as Ts65Dn) mice (The Jackson Laboratory, Bar Harbor, Maine; stock #: 001924). For ex vivo slice experiments to label SST-positive MCs, Ts65Dn mice were crossed with *Tg(Gad1-EGFP)98Agmo/J*, also known as GAD67-GFP X98 or GFP-X98 (Jackson Laboratory). To label PV-INs, Ts65Dn mice were crossed with *C57BL/6-Tg(Pvalb-tdTomato)15Gfng/J* (*Pvalb-tdTomato*, Jackson Laboratory). Mice used in this study were of both sexes.

## In vivo LFP and juxtacellular recordings

Ts or Eu mice were anesthetized with 15% urethane (1.5 g/kg in physiological solution) and placed on a stereotaxic apparatus. The body temperature was constantly monitored and kept at 37°C with a heating blanket. To ensure a deep and constant level of anesthesia, vibrissae movement, eyelid reflex, response to tail, and toe pinching were visually controlled before and during the surgery. A local lidocaine injection was performed over the cranial area of interest and, after a few minutes, a longitudinal incision was performed to expose the skull. Two small cranial windows (<1 mm diameter) were opened at at 2.5 mm from bregma and ±0.5 mm lateral to sagittal sinus (corresponding to the frontal lobe) carefully avoiding any damage to the main vessels while keeping the surface of the brain moist with the normal HEPES-buffered artificial cerebrospinal fluid. Pipettes used to record LFP had 1–2 MΩ resistance while those used for juxtacellular patch-clamp recordings typically had 5–7 MΩ resistance. LFP and patch electrodes were pulled from borosilicate glass capillaries. Signals were amplified with a Multiclamp 700B patch-clamp amplifier (Molecular Devices), sampled at 20 KHz and filtered online at 10 KHz. Signals were digitized with a Digidata 1440A and acquired, using the pClamp 10 software package (Molecular Devices).

## Preparation of acute slices for electrophysiology

In order to record intrinsic and synaptic properties of L2/3 neurons of mPFC, we prepared acute cortical slices from the described mouse lines. For these experiments, we used slices cut in the coronal plane (300–350 μm thick). Animals were deeply anesthetized with saturating isofluorane (Vetflurane, Virbac) and immediately decapitated. The brain was then quickly removed and immersed in the cutting choline-based solution, containing the following (in mM): 126 choline chloride, 16 glucose, 26 $NaHCO_3$, 2.5 KCl, 1.25 $NaH_2PO_4$, 7 $MgSO_4$, 0.5 $CaCl_2$, cooled to 4°C and equilibrated with a 95–5% $O_2$-$CO_2$ gas mixture. Slices were cut with a vibratome (Leica VT1200S) in cutting solution and then incubated in oxygenated artificial cerebrospinal fluid (aCSF) composed of (in mM): 126 NaCl, 20 glucose, 26 $NaHCO_3$, 2.5 KCl, 1.25 $NaH_2PO_4$, 1 $MgSO_4$, 2 $CaCl_2$ (pH 7.35, 310-320mOsm/L), initially at 34°C for 30 min, and subsequently at room temperature, before being transferred to the recording chamber where recordings were obtained at 30–32°C.

## Slice electrophysiology

Whole-cell patch-clamp recordings were performed in L2/3 of the medial prefrontal cortex (mPFC) neurons. Inhibitory PV-expressing interneurons, labeled with TdTomato in Ts65Dn mice crossed with *Pvalb-tdTomato* mice and Martinotti cells, labeled with GFP in Ts65Dn crossed with GFP-X98 mice, were identified using LED illumination (OptoLED, Cairn Research, Faversham, UK). Excitatory pyramidal neurons (PNs) were visually identified using infrared video microscopy, as cells lacking expression of fluorescent proteins and with somatas exhibiting the classical pyramidal shape. Accordingly, when depolarized with DC current pulses PNs exhibited a typical firing pattern of regular-spiking cells. We used different intracellular solutions depending on the type of experiment and the nature of the responses we wanted to assess. To study passive properties, intrinsic excitability, AP waveform and glutamatergic spontaneous transmission, electrodes were filled with an intracellular solution containing (in mM): 127 K-gluconate, 6 KCl, 10 Hepes, 1 EGTA, 2 MgCl2, 4 Mg-ATP, 0.3 Na-GTP; pH adjusted to 7.3 with KOH; 290–300 mOsm. The estimated reversal potential for chloride ($E_{Cl}$) was approximately −69 mV based on the Nernst equation. To measure GABAergic currents elicited by perisomatic-targeting interneurons, PNs were patched using an intracellular solution containing (in mM): 65 K-gluconate, 70 KCl, 10 Hepes, 1 EGTA, 2 MgCl2, 4 Mg-ATP, 0.3 Na-GTP; pH adjusted to 7.3 with KOH; 290–300 mOsm (the estimated ECl was approximately −16 mV based on the Nernst equation). For distal dendritic uIPSCs, we used a cesium-based solution containing (in mM): 145 CsCl, 10 Hepes, 1 EGTA, 0.1 CaCl2, 2 MgCl2, 4.6 Mg-ATP, 0.4 Na-GTP, 5 QX314-Cl; pH adjusted to 7.3 with CsOH; 290–300 mOsm. Under these recording conditions, activation of GABA$_A$ receptors resulted in inward currents at a holding potential (Vh) of −70 mV. Voltage values were not corrected for liquid junction potential. Patch electrodes were pulled from borosilicate glass capillaries and had a typical tip resistance of 2–3 MΩ. Signals were amplified with a Multiclamp 700B patch-clamp amplifier (Molecular Devices), sampled at 20–50 KHz and filtered at 4 KHz (for voltage-clamp experiments) and 10 KHz (for current-clamp experiments). Signals were digitized with a Digidata 1440A and acquired, using the pClamp 10 software package (Molecular Devices).

*For paired recordings*, unitary synaptic responses were elicited in voltage-clamp mode by brief somatic depolarizing steps (−70 to 0 mV, 1–2 ms) evoking action currents in presynaptic cells. Neurons were held at −70 mV and a train of 5 presynaptic spikes at 50 Hz was applied.

## α5IA

3-(5-methylisoxazol-3-yl)−6-[(1-methyl-1,2,3-triazol-4-yl)methyloxy]−1, 2, 4-triazolo[3, 4-a]phthalazine also named L-822179, IUPAR/BPS 4095 or PubChem CID 6918451 was synthesized by Orga-Link SARL (Magny-les-Hameaux, France), according to *Sternfeld et al., 2004* as in *Braudeau et al., 2011*.

## Immunohistochemistry

Parvalbumin, SST and GFP staining were performed on 20–50 μm-thick slices. Briefly, mice were perfused with 0.9% NaCl solution containing Heparin and 4% paraformaldehyde (PFA). Brains were cryo-protected by placing them overnight in 30% sucrose solution and then frozen in Isopentane at a temperature <-50°C. Brains were sliced with a freezing microtome (ThermoFisher HM450). Permeabilization in a blocking solution of PBT with 0.3% Triton and 10% Normal Goat Serum was done at room temperature for 2 hr. Slices were then incubated overnight (4°C) in the same blocking solution containing the primary rabbit anti-PV antibody (1:1000; Thermo Scientific) and mouse anti-SST antibody (1:250; Santa Cruz Biotechnologies). Slices were then rinsed three times in PBS (10 min each) at room temperature and incubated with goat anti-rabbit and a goat anti-mouse antibody (1:500; Jackson IR) coupled to Alexa-488 or 633 for 3.5 hr at room temperature. Slices were then rinsed three times in PBS (10 min each) at room temperature and coverslipped in mounting medium (Fluoromount, Sigma Aldrich F4680). Immunofluorescence was then observed with a slide scanner (Zeiss, Axio Scan.Z1).

## Morphological reconstruction

Biocytin Fills: To reliably reconstruct the fine axonal branches of cortical neurons, dedicated experiments were performed following the classical avidin-biotin-peroxidase method. Biocytin (Sigma) was added to the intracellular solution at a high concentration (5–10 mg/ml), which required extensive

sonication. At the end of recordings, the patch pipette was removed carefully until obtaining an inside out patch. The slice was then left in the recording chamber for at least further 5–10 min to allow further diffusion. Slices were then fixed with 4% paraformaldehyde in phosphate buffer saline (PBS, Sigma) for at least 48 hr. Following fixation, slices were incubated with the avidin-biotin complex (Vector Labs) and a high concentration of detergent (Triton-X100, 5%) for at least two days before staining with 3,3'Diaminobenzidine (DAB, AbCam). Cells were then reconstructed and cortical layers delimited using Neurolucida 7 (MBF Bioscience) and the most up to date mouse atlas (Allen Institute).

## Data analysis

Electrophysiological and statistical analysis was performed using built-in and custom-written routines made for Igor Pro (WaveMetrics, Lake Oswego, OR, USA), MATLAB R2017b 9.3.0.713579 Natick, Massachusetts: The MathWorks Inc; Origin (Pro) 2016 OriginLab Corporation, Northampton, MA, USA; Prism version 7.00 for Windows, GraphPad Software, La Jolla California USA; and Python Software Foundation. Python Language Reference, version 3.6, available at http://www.python.org.

## Analysis of in vivo recordings

Traces obtained from juxtacellular recording were high pass filtered (cutoff: 5 Hz) and spikes were detected based on threshold = 1.5 mV. Spike rate was estimated as the total number of spikes detected divided by the total duration of the recording. The peak time ($t_{peak}$) corresponding to each detected action potential was used to select the segment of LFP between: [$t_{peak}$-100 ms to $t_{peak}$+100 ms]. The instantaneous phase was estimated using Hilbert transform on decimated LFP and subsequently used to estimate phase locking. For LFP analysis, extracellular potentials were down-sampled (1 kHz) and low- pass filtered (cutoff frequency, 100 Hz). Power spectra were generated using a Hann window (window length: 4096 points, 50% overlap).

Phase locking was determined using pairwise phase consistency (PPC) estimation, defined as:

$$PPC_f = \frac{2\left(\sum_{i=1}^{N-1}\sum_{j=i+1}^{N}\left(cos(\theta_i)cos(\theta_j) + sin(\theta_i)sin(\theta_j)\right)\right)}{N.(N-1)}$$

Where N is the total number of action potentials and $\theta_i$ is the phase of the $i^{th}$ spike and $\theta_j$ the $j^{th}$.

## UP and DOWN states detection

Cortical states were detected as described elsewhere (*Ruiz-Mejias et al., 2011*). Briefly, filtered and decimated LFP was used to calculate UP and DOWN state likelihood decision (or evidence) variable ($S_{comb}$) based on low frequency (<4 Hz) oscillation phase and high frequencies (20–100 Hz) composition of the LFP. To determine the thresholds to detect different cortical states (UP, Intermediate, DOWN) states, the distribution of the combined evidence variable, $S_{comb}$, was fitted by a mixture of three Gaussians, each representing their corresponding cortical state, UP (highest level of the signal), Intermediate (Intermediate level) and DOWN (lowest level of the signal). Periods of the combined signal, $S_{comb}$, that were above the UP threshold, $\mu_{UP-LFP} - 3 * \sigma_{UP-LFP}$, were considered the periods of UP states. Similarly, the periods below the DOWN threshold, $\mu_{DOWN-LFP} - 3 * \sigma_{DOWN-LFP}$, were considered the periods of DOWN states (means and variances of the Gaussians are represented as $\mu_{UP}$, $\mu_{DOWN}$, and $\sigma_{UP}$, $\sigma_{DOWN}$ for the up and down cortical states, respectively). Periods of UP and DOWN states were refined further by putting constraints on the interval between two states and duration of a state. Minimum interval between two states and duration of a state were set 50 and 70 ms, respectively.

## Analysis of slices electrophysiological recordings

Input resistance ($R_i$) was estimated as the slope of the current to voltage relationship obtained with upon the injection of −25, 0 and 25 pA to a cell kept at resting potential. Membrane time constant was estimated fitting the time course of $V_{memb}$ after the injection of a 2 s, −25 pA current step.

We used protocols of increasing steps of current injection (−50 to 500 pA in steps of 25 or 50 pA and 2 s duration). Action potentials were detected using a threshold based routine. Threshold was

set at 0 mV. Firing dynamics was evaluated fitting AP frequency versus current relationship (F-I curve) for somatic current injections from individual cells to a logarithmic function:

$$f(I) = gain * ln\left(\frac{I}{rheobase}\right)$$

Where $I$ is the amount of injected current, the parameter gain represents the gain of the system and rheobase represents the minimal amount of current required to trigger an action potential. The first action potential evoked at rheobase was taken to measure amplitude, width and threshold. Threshold was considered as the potential at which dV/dt reached 10 mV/ms; amplitude was the difference between peak amplitude and threshold and half-width was the time interval between rise and decay phase measured at 50% of amplitude.

Action potential threshold was defined as the membrane potential value ($V_m$) at which dV/dt becomes larger than 10 mV/ms. Action potential amplitude was defined as:

$$Amp = Amp_{peak} - threshold,$$

where $Amp_{peak}$ was the AP peak potential. Action potential width was measured at 50% of amplitude.

## Statistical analysis

Normal distribution of samples was systematically assessed (Shapiro-Wilkinson normality test). Normal distributed samples were statistically compared using two-tailed Student's $t$ test unless otherwise stated. When data distribution was not normal we used two-tailed Mann Whitney U-test. Compiled data are reported and presented as whisker box plots the upper and lower whiskers representing the 90th and 10th percentiles, respectively, and the upper and lower boxes representing the 75th and 25th percentiles, respectively, and the horizontal line representing the median or the mean $\pm$ s.e.m., with single data points plotted. Differences were considered significant if $p<0.05$ (*$p<0.05$, **$p<0.01$, ***$p<0.001$).

## Dendritic inhibition

Evaluation of synaptic efficiency in the dendritic inhibitory loop composed by MCs and PNs.

## Perisomatic inhibition

Evaluation of synaptic efficiency in the perisomatic inhibitory loop composed by PV cells and PNs.
Ts65Dn in vivo activity.

# Acknowledgements

We thank Vikaas Sohal, Nelson Rebola and Maria del Mar Dierssen Sotos for critically reading this manuscript and Michele Giugliano for insightful discussions. We also thank the ICM technical facilities PHENO-ICMICE and iGENSEQ.

This work was supported by 'Investissements d'avenir' ANR-10-IAIHU-06, BBT-MOCONET; Agence Nationale de la Recherche (ANR-12-EMMA-0010 ; ANR-13-BSV4-0015-01; ANR-16-CE16-0007-02; ANR-17-CE16-0026-01; ANR-18-CE16-0001-01), Fondation Recherche Médicale (Equipe FRM DEQ20150331684 and EQU201903007860), NARSAD independent investigator grant, École des Neurosciences de Paris Ile-de-France and Fondation Lejeune (#1790). All animal work was conducted at the PHENO-ICMice facility. The PHENO-ICMICE Core is supported by 2 'Investissements d'avenir' (ANR-10- IAIHU-06 and ANR-11-INBS-0011-NeurATRIS) and the 'Fondation pour la Recherche Médicale'.

## Additional information

### Funding

| Funder | Grant reference number | Author |
|---|---|---|
| Fondation Jérôme Lejeune | #1790 | Javier Zorrilla de San Martin |

| ICM - Institut du Cerveau | BBT-MOCONET | Marie-Claude Potier<br>Alberto Bacci |
|---|---|---|
| Fondation Recherche Medicale - Equipe FRM | DEQ20150331684 | Alberto Bacci |
| National Alliance for Research on Schizophrenia and Depression | | Alberto Bacci |
| Fondation Recherche Medicale - Equipe FRM | EQU201903007860 | Alberto Bacci |
| Agence Nationale de la Recherche - ANR | ANR-13-BSV4-0015-01 | Alberto Bacci |
| Agence Nationale de la Recherche - ANR | ANR-17-CE16-0026-01 | Alberto Bacci |
| Agence Nationale de la Recherche - ANR | ANR-18-CE16-0001-01 | Alberto Bacci |
| Agence Nationale de la Recherche - ANR | ANR-10-IAIHU-06 | Marie-Claude Potier<br>Alberto Bacci |
| Agence Nationale de la Recherche - ANR | ANR-12-EMMA-0010 | Marie-Claude Potier |
| Agence Nationale de la Recherche - ANR | ANR-16-CE16-0007-02 | Marie-Claude Potier |

The funders had no role in study design, data collection and interpretation, or the decision to submit the work for publication.

## Author contributions

Javier Zorrilla de San Martin, Conceptualization, Data curation, Software, Formal analysis, Supervision, Funding acquisition, Validation, Investigation, Visualization, Methodology, Writing - original draft, Project administration, Writing - review and editing; Cristina Donato, Formal analysis, Investigation, Methodology, Project administration, Writing - review and editing; Jérémy Peixoto, Formal analysis, Investigation, Writing - review and editing; Andrea Aguirre, Formal analysis, Investigation, Methodology, Writing - review and editing; Vikash Choudhary, Software, Formal analysis, Methodology, Writing - review and editing; Angela Michela De Stasi, Joana Lourenço, Investigation, Methodology, Writing - review and editing; Marie-Claude Potier, Conceptualization, Resources, Supervision, Funding acquisition, Validation, Visualization, Methodology, Writing - original draft, Project administration, Writing - review and editing; Alberto Bacci, Conceptualization, Resources, Software, Formal analysis, Supervision, Funding acquisition, Validation, Visualization, Writing - original draft, Project administration, Writing - review and editing

## Author ORCIDs

Javier Zorrilla de San Martin (iD) https://orcid.org/0000-0003-2848-7482
Cristina Donato (iD) https://orcid.org/0000-0003-4078-0745
Jérémy Peixoto (iD) https://orcid.org/0000-0002-6814-9747
Andrea Aguirre (iD) https://orcid.org/0000-0003-1176-8747
Joana Lourenço (iD) http://orcid.org/0000-0001-5550-9291
Marie-Claude Potier (iD) https://orcid.org/0000-0003-2462-7150
Alberto Bacci (iD) https://orcid.org/0000-0002-3355-5892

## Ethics

Animal experimentation: Experimental procedures followed National and European guidelines, and have been approved by the authors' institutional review boards (French Ministry of Research and Innovation, APAFIS#2599-2015110414316981v21). Every effort was made to minimize suffering.

## Decision letter and Author response

Decision letter https://doi.org/10.7554/eLife.58731.sa1

Author response https://doi.org/10.7554/eLife.58731.sa2

## Additional files

### Supplementary files
• Transparent reporting form

### Data availability

Source data files have been provided for: Figure 1, Figure 1–figure supplement 2, Figure 1–figure supplement 3, Figure 2, Figure 2–figure supplement 1, Figure 2–figure supplement 2, Figure 3, Figure 3–figure supplement 1, Figure 4, Figure 4–figure supplement 1 and Figure 5.

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
