## [Decision Letter]

**Acceptance summary:**

Down Syndrome is a developmental disorder due to a chromosomal abnormality which can be modeled in mice. Recent work has suggested that the intellectual disability results from excess inhibition in brain circuits. The authors use elegant physiological methods to provide direct evidence for the proposed over-inhibition in a mouse model of Down Syndrome and identify specific changes differently affecting two well studied classes of GABAergic interneurons.

**Decision letter after peer review:**

Thank you for submitting your article "Alterations of specific cortical GABAergic circuits underlie abnormal network activity in a mouse model of Down syndrome" for consideration by *eLife*. Your article has been reviewed by three peer reviewers, including Sacha B Nelson as the Reviewing Editor and Reviewer #1, and the evaluation has been overseen by Gary Westbrook as the Senior Editor. The following individual involved in review of your submission has agreed to reveal their identity: Josef Bischofberger (Reviewer #2). The reviewers have discussed the reviews with one another and the Reviewing Editor has drafted this decision to help you prepare a revised submission.

Summary:

The concept of "E/I balance" has dominated discussions of the etiology of developmental disorders involving intellectual disability and Autism Spectrum Disorders. But many have rightly complained that this concept is poorly defined and poorly specified. The disorder Down Syndrome (DS) is especially interesting in this regard, both because it is the most common genetic syndrome causing intellectual disability and because pharmacological studies in mice have argued for enhanced inhibition as an important cause. The present study presents the first direct evidence for enhanced inhibition in mice that model the trisomy underlying DS. The authors use a rigorous approach involving paired recordings between identified types of neurons and identify specific synaptic and intrinsic changes in two important subpopulations of cortical interneurons; the Sst-positive Martinotti cells, and Pvalb-expressing basket cells.

The authors should make textual changes to address the key points raised by the three reviewers. Specifically,

1) Clarify and better discuss the limitations and caveats to the experiments involving pharmacological manipulation of enhanced dendritic inhibition.

2) Provide further analyses, clarification and discussion of the alterations in Pvalb-neuron excitability, along the lines suggested by reviewers #2 and #3 below.

3) Enhance the discussion of the final figure to better integrate these results into the rest of the paper.

The logic of the requested textual revisions is explained in the concerns of the reviewers and so those are included below.

Revisions expected in follow-up work:

If available, follow up recordings in older animals would add to the impact of the work as would follow up pharmacology studies.

Reviewer #1:

The authors provide direct evidence for the proposed over-inhibition in a mouse model of Down Syndrome and identify specific changes differently affecting two well studied classes of GABAergic interneurons. The results are robust and well-illustrated and the experiments are rigorous and well designed.

I have mainly minor suggestions for improving the manuscript. The most significant concern is that the English usage would benefit from careful editing.

Reviewer #2:

The new paper “Alterations of specific cortical GABAergic circuits underlie abnormal network activity in a mouse model of Down Syndrome” investigates how altered synaptic transmission in prefrontal cortex (PFC) might impact on cortical network activity in Down Syndrome. Using elegant electrophysiological paired recordings of synaptically coupled Martinotti-cells, PV-interneurons and pyramidal cells, the authors nicely show that dendritic inhibition is enhanced in the PFC. By contrast, peri-somatic inhibition is relatively normal. Furthermore, it is claimed that the excitability of PV-basket cells is increased affecting cortical γ-oscillations in anesthetized mice. To support these later claims, some more analysis is necessary.

1) Increased dendritic inhibition. This first part is very convincing in general. However, I was wondering about the 5-fold increase in synaptic charge, while peak amplitudes are only 3-times larger (Figure 1C). Is the decay time course slower in TS-mice? Furthermore, I could not find any information about the used GABA receptor modulator (a5IA). Is it L-822179 (PubChem CID 6918451)? There are many inverse agonists/modulators of GABAa5 receptors available. It is important that the authors precisely specify which drug and from what source. Finally, the authors should refer correctly to Schulz et al., 2019 and discuss that increased dendritic inhibition via GABAa5 receptors was also found in CA1 pyramidal cells of Ts65Dn mice.

2) PV-firing. Firstly, I am surprised by the control data of PV-cells in Figure 4. It seems that there is no defined rheobase in Figure 4B in contrast to the traces shown in Figure 4A black. Did the authors record form separate populations of PV-interneurons which distorts the average FI-curve? It would be important to analyze the FI-curves of different cells separately (as for example performed in Schulz et al., 2019). This will allow to calculate mean values (and medians) for rheobase and gain. A histogram (or cumulative distribution) of rheobase and gain will show the presence of different populations of cells in control and in Ts65Dn. To do so, the formula in subsection “Analysis of slices electrophysiological recordings” should be replaced by a more appropriate function, which includes a rheobase and without a base firing which is zero anyway (for example F = gain*ln(I/rheobase)). Second, the mice for slices are rather young (P18-P25). In adult animals, cortical PV-tdTom cells appear to show a mean Rin of 118 MOhm, an AP half duration of 0.5 ms and a taum of 7.2 ms (Kaiser et al., 2016). The control animals in the present study show consistently larger values. The values in Ts65Dn mice are further increased. Is it possible that PV cells of TS animals are just delayed in development? A first indication could be a correlative analysis plotting AP with, Rin, rheobase, gain and max firing frequency against animal age (P18-P25) to see whether there are significant correlations. The best would be slice recordings using older animals (> 4 weeks), to see whether the differences disappear.

3) in vivo field potentials. The field potential analysis is only loosely connected to the rest of the paper. How do we know whether “alterations in cortical GABA circuits underlie the abnormal network activity” and increased γ-power in the in vivo LFPs? Furthermore, the authors discuss these data in the context of temporal coding and cognition. The slow waves during urethane anesthesia (1-4Hz) resembles slow wave sleep, which is relevant for memory consolidation. The authors should discuss the new paper by Alemany-Gonzalez et al., 2020 (PNAS 117:11788-11798), which show disturbed slow wave sleep oscillations in Ts65Dn mice. Similar to the present study they find increased γ power in PFC during slow wave sleep (Alemany-Gonzalez, Figure 2B), which could be related to the well-known sleep disturbances in DS patients.

Reviewer #3:

The authors present a clearly written, technically sound and very interesting study in which they demonstrate substantial changes in the prefrontal cortex inhibitory circuitry of a well-characterized transgenic Down Syndrome model. They show an aberrant dendritic inhibitory circuitry caused by an increased in synaptic transmission from and to Martinotti cells in the medial prefrontal cortex and changes in excitability of perisomatic inhibitory interneurons without apparent changes in synaptic properties. Finally, they demonstrate a strong decrease in principal cell activity on the medial prefrontal cortex of urethane anesthetized Ts mice and increase coupling to γ oscillations.

I find the study interesting and suitable for publication in *eLife*. It gives a detailed description of the inhibitory circuitry in the Ts mice and provides new in vivo evidence supporting the theory of over-inhibition as a main cause for cognitive deficits in DS. Nevertheless, there are some issues I think should be addressed first.

1) There is a mismatch between the age of animals used for in vitro and in vivo recordings. Down syndrome is a developmental disease, and it could be that interneuron development might take more time than in WT animals. PV positive interneurons are known to develop in the hippocampus during the first 4 postnatal weeks, bringing the question if the abnormalities observed in this study remain during adulthood. I suggest performing some of the main in vitro experiments for parvalbumin and somatostatin interneurons in slices from adult animals.

2) Another concern relates to the final conclusion of the paper, which correlates the in vitro observed abnormalities in inhibition with the decreased activity of principal cells. Although this might be the most rational assumption, I would appreciate evidence to make this hypothesis more likely, considering that DS mice also show changes in spine density, NMDA receptor expression in principal cells and increase threshold for firing as shown in this study. This could be performed by:

– Evaluating the effect of α5IA injection on principal cell activity in Eu and Ts mice.

– Performing recordings from identified interneurons in vivo in Eu and Ts mice.

– Test the effects of specific inhibition of parvalbumin or somatostatin interneurons on the activity of pyramidal cells in Eu and Ts mice.

A more comprehensive discussion addressing these issues might also be sufficient in case experiments are not feasible in a reasonable time frame.

3) The concentration of α5IA of 100 nm seems not to provide the highest specificity of the drug, having a milder but significant effect in receptors containing the α3 and α1 subunits (Dawson et al., 2005). Considering the general importance that this strong and very specific effect may have for the community, and the sparse literature using this drug in vitro I would appreciate a better characterization of this particular experiment.

– Does α5IA changes the kinetics of the GABAergic response?

– Are PV-mediated IPSCs affected by this concentration of α5IA?

– Does the α5IA dependent response shows voltage-dependency as has been reported (Schultz et al., 2019)

---

## [Author Response]

Reviewer #1:The authors provide direct evidence for the proposed over-inhibition in a mouse model of Down Syndrome and identify specific changes differently affecting two well studied classes of GABAergic interneurons. The results are robust and well-illustrated and the experiments are rigorous and well designed.I have mainly minor suggestions for improving the manuscript. The most significant concern is that the English usage would benefit from careful editing.

We have revised the English language throughout the manuscript and we asked a native speaker to read it. We hope the revised text is much clearer than the previous version.

Reviewer #2:The new paper “Alterations of specific cortical GABAergic circuits underlie abnormal network activity in a mouse model of Down Syndrome” investigates how altered synaptic transmission in prefrontal cortex (PFC) might impact on cortical network activity in Down Syndrome. Using elegant electrophysiological paired recordings of synaptically coupled Martinotti-cells, PV-interneurons and pyramidal cells, the authors nicely show that dendritic inhibition is enhanced in the PFC. By contrast, peri-somatic inhibition is relatively normal. Furthermore, it is claimed that the excitability of PV-basket cells is increased affecting cortical γ-oscillations in anesthetized mice. To support these later claims, some more analysis is necessary.1) Increased dendritic inhibition. This first part is very convincing in general. However, I was wondering about the 5-fold increase in synaptic charge, while peak amplitudes are only 3-times larger (Figure 1C). Is the decay time course slower in TS-mice?

Whereas the amplitude was calculated on the first synaptic response, the charge was measured over the entire train. We have measured the kinetics of MC-PN uIPSCs. We found no significant differences in isolated uIPSC decay time constant in Eu vs. Ts mice (see the new Figure 1—figure supplement 2). We found a significant increase of uIPSC rise-times in Eu vs Ts. This could be ascribed to a more distal location of MC-PN synapses on dendrites of Ts mice. Although this change could in principle account for, at least in part, an increase in synaptic charge, the magnitude of this change is unlikely to account for a 5-fold increase.

We speculate that a 5-fold increase of synaptic charge is due to one or a combination of different non-linearities that are dynamically emerging during a spike train: (i) recruitment of peri- or extrasynaptic GABA_A_Rs due to enhanced release of GABA; (ii) changes in postsynaptic receptors occupation and desensitization. Complex interplays between these factors occur at distal dendrites, where we have poor voltage control due to space-clamping constraints. Future studies will be necessary to pinpoint the exact biophysical and anatomical alterations underlying the prominent increase of dendritic inhibition operated by MCs in DS.

We have generated a new figure supplement to Figure 1 (Figure 1—figure supplement 2), detailing this finding. We have described (in subsection “Synaptic enhancement of dendritic inhibition in DS”) and integrated this new analysis to the discussion in the revised manuscript.

Furthermore, I could not find any information about the used GABA receptor modulator (a5IA). Is it L-822179 (PubChem CID 6918451)? There are many inverse agonists/modulators of GABAa5 receptors available. It is important that the authors precisely specify which drug and from what source.

The α5IA, an inverse antagonist of α5-containing GABAA receptors used in our study corresponds to the molecule ID:4095 (IUPHAR/BPS), also named L-822179, or CID 6918451 in PubChem. This has been detailed in the Materials and methods section of the revised manuscript, with the appropriate references.

Finally, the authors should refer correctly to Schulz et al., 2019 and discuss that increased dendritic inhibition via GABAa5 receptors was also found in CA1 pyramidal cells of Ts65Dn mice.

We thank the reviewer for pointing this out. We have correctly addressed this paper in the revised manuscript (Discussions section).

2) PV-firing. Firstly, I am surprised by the control data of PV-cells in Figure 4. It seems that there is no defined rheobase in Figure 4B in contrast to the traces shown in Figure 4A black. Did the authors record form separate populations of PV-interneurons which distorts the average FI-curve? It would be important to analyze the FI-curves of different cells separately (as for example performed in Schulz et al., 2019). This will allow to calculate mean values (and medians) for rheobase and gain. A histogram (or cumulative distribution) of rheobase and gain will show the presence of different populations of cells in control and in Ts65Dn. To do so, the formula in subsection “Analysis of slices electrophysiological recordings” should be replaced by a more appropriate function, which includes a rheobase and without a base firing which is zero anyway (for example F = gain*ln(I/rheobase)).

We have re-analyzed f-i curves as suggested by the reviewer. The new Figure 4 includes two representative examples of f-i curves in the two genotypes, fitted with the function suggested by the reviewer (panel b). Moreover, population data relative to rheobase and gain are shown in box-plot panels (left panel in c). This new analysis is consistent with a strong alteration of PV-cell excitability and firing dynamics in Ts mice. We have detailed this new analysis in the revised manuscript (subsection “Excitability of PV cells, and not their perisomatic control of PNs, is strongly altered in Ts mice”).

Second, the mice for slices are rather young (P18-P25). In adult animals, cortical PV-tdTom cells appear to show a mean Rin of 118 MOhm, an AP half duration of 0.5 ms and a taum of 7.2 ms (Kaiser et al., 2016). The control animals in the present study show consistently larger values. The values in Ts65Dn mice are further increased. Is it possible that PV cells of TS animals are just delayed in development? A first indication could be a correlative analysis plotting AP with, Rin, rheobase, gain and max firing frequency against animal age (P18-P25) to see whether there are significant correlations. The best would be slice recordings using older animals (> 4 weeks), to see whether the differences disappear.

The reviewer raises an excellent point. Indeed, active and passive properties of PV cells of Ts mice are consistent with an immature phenotype (Okaty et al., 2009). We performed a correlation analysis of the passive properties and excitability with age as suggested. In Eu mice, we found that input resistance, tau membrane and AP half width change along this period of development into values similar to those reported by Okaty et al., 2009 and Kaiser et al., 2016. We have added a figure supplement to Figure 4 illustrating this developmental profile (see Figure 4—figure supplement 2).

Importantly, if we confine our analysis in the older time window of our recordings (P23-25), we found that differences in AP width, input resistance and membrane time constant remain starkly different. We include Author response image 1 and Author response table 1 illustrating this finding at this age. We also include the values of the Kaiser paper obtained in the somatosensory cortex of adult mice. Overall, this analysis excludes potential effect of a bias in the age composition of our sample and validates our interpretation.

**Author response image 1. sa2fig1:** 

**Author response table 1. resptable1:** 

	Eu (n = 8) P23-25	Ts (n = 10) P23-25	P value*	Kaiser et al. 2016^§^ P45-90
Ri (MΩ)	94.6 ± 6.7	200.3 ± 45.3	p = 0.0034	119 ± 4.0
τm (ms)	6.9 ± 0.9	25.7 ± 4.2	p = 0.0004	7.2 ± 0.2
AP width (ms)	0.66 ± 0.03	1.44 ± 0.20	p = 0.0026	0.5 ± 0.01

*Two-sided Mann Whitney U test.

^§^The age of mice in this paper was P45-90 and measurements were done in S1.

However, the reviewer is right in pointing that in vivo recordings were obtained in adult animals. Moreover, we agree with the reviewer that it is important to determine whether this phenotype is transitory or if it lasts into adulthood. We shared this view, and indeed, we had started to record from adult animals, but we were forced to suspend all experiments and sacrifice our mouse colonies due to the current pandemic crisis in France. We are in the process of obtaining new colonies specifically for these experiments. In the following months, we will test the hypothesis of delayed development to be exploited in a future study that will include cross-disciplinary approaches.

3) in vivo field potentials. The field potential analysis is only loosely connected to the rest of the paper. How do we know whether “alterations in cortical GABA circuits underlie the abnormal network activity” and increased γ-power in the in vivo LFPs? Furthermore, the authors discuss these data in the context of temporal coding and cognition. The slow waves during urethane anesthesia (1-4Hz) resembles slow wave sleep, which is relevant for memory consolidation. The authors should discuss the new paper by Alemany-Gonzalez et al., 2020 (PNAS 117:11788-11798), which show disturbed slow wave sleep oscillations in Ts65Dn mice. Similar to the present study they find increased γ power in PFC during slow wave sleep (Alemany-Gonzalez, Figure 2B), which could be related to the well-known sleep disturbances in DS patients.

The reviewer raises a series of good points. We cannot directly link the inhibitory circuit-specific alterations that we detect in slices with the increased synchronization of β-γ-activity we see in vivo. However, a large body of literature indicates that oscillations in this frequency range strongly depends on the activity of PV cells (Buzsaki and Wang, 2012; Sohal et al., 2009; Cardin et al., 2009). More recently, also SST interneurons were shown to be involved in low frequency (30 Hz) neocortical γ-oscillations (Veit et al., 2017). Here, we find evidence for over-inhibition in Ts mice in vivo, accompanied by LFP power increases in β-γ-frequency ranges. These observations are consistent with increased interneuron activity (Cardin et al., 2009; Sohal et al., 2009; Atallah et al., 2012). Future studies will be necessary to identify the specific roles played by PV interneurons and MCs in controlling PN spiking in vivo its temporal correlation with network oscillations in Ts mice. These studies will be lengthy, as they will require controlling PV or MC firing using pharmacogenetics. We have better discussed this in the revised manuscript (Discussion section).

We agree with the reviewer that our results are more pertinent with slow-wave sleep and memory consolidation. We have revised our interpretation throughout the manuscript. We have also cited the recent report by Alemany-González et al., 2020, which showed increased low β- and γ-band during sleep and disrupted functional connectivity between the PFC and the hippocampus during learning in Ts65Dn mice.

Reviewer #3:The authors present a clearly written, technically sound and very interesting study in which they demonstrate substantial changes in the prefrontal cortex inhibitory circuitry of a well-characterized transgenic Down Syndrome model. They show an aberrant dendritic inhibitory circuitry caused by an increased in synaptic transmission from and to Martinotti cells in the medial prefrontal cortex and changes in excitability of perisomatic inhibitory interneurons without apparent changes in synaptic properties. Finally, they demonstrate a strong decrease in principal cell activity on the medial prefrontal cortex of urethane anesthetized Ts mice and increase coupling to γ oscillations.I find the study interesting and suitable for publication in eLife. It gives a detailed description of the inhibitory circuitry in the Ts mice and provides new in vivo evidence supporting the theory of over-inhibition as a main cause for cognitive deficits in DS. Nevertheless, there are some issues I think should be addressed first.1) There is a mismatch between the age of animals used for in vitro and in vivo recordings. Down syndrome is a developmental disease, and it could be that interneuron development might take more time than in WT animals. PV positive interneurons are known to develop in the hippocampus during the first 4 postnatal weeks, bringing the question if the abnormalities observed in this study remain during adulthood. I suggest performing some of the main in vitro experiments for parvalbumin and somatostatin interneurons in slices from adult animals.

This is an excellent point that was similarly raised by reviewer #2. Please see the response to his point 2. We have re-analyzed our data, indicating that active and passive properties of PV cells in control animals were similar to those recorded in adults (Okaty et al., 2009 and Kaiser et al., 2016; see Author response table 1). However, we agree with the reviewer that it is important to determine whether the phenotypes observed are transitory or if they last throughout adulthood. We shared this view, and indeed, we had started to record from adult animals (X98 crossed with Ts65Dn and PvalbTdTomato crossed with Ts65Dn), but we were forced to suspend all experiments and sacrifice the mice due to the current pandemic crisis in France. We are in the process of obtaining new colonies specifically for these experiments in >P70 mice. In the following months, we will be able to test the hypothesis of delayed development of PV cells. Results will be exploited in a future study that will include cross-disciplinary approaches. We have discussed this possibility in the revised manuscript.

2) Another concern relates to the final conclusion of the paper, which correlates the in vitro observed abnormalities in inhibition with the decreased activity of principal cells. Although this might be the most rational assumption, I would appreciate evidence to make this hypothesis more likely, considering that DS mice also show changes in spine density, NMDA receptor expression in principal cells and increase threshold for firing as shown in this study. This could be performed by:– Evaluating the effect of α5IA injection on principal cell activity in Eu and Ts mice.– Performing recordings from identified interneurons in vivo in Eu and Ts mice.– Test the effects of specific inhibition of parvalbumin or somatostatin interneurons on the activity of pyramidal cells in Eu and Ts mice.A more comprehensive discussion addressing these issues might also be sufficient in case experiments are not feasible in a reasonable time frame.

The reviewer raises an excellent point. Please see our response to point 3 of Rev. #2, who raised a similar concern. In particular, we agree with the reviewer that what he/she proposes would be excellent experiments to complement and corroborate our conclusions. However, as detailed above, we are presently in a position that we cannot start even the easiest experiments before six months. In particular, 2-photon assisted recordings from identified interneurons cannot be performed in the mPFC, as it is too deep to allow 2P imaging. We would need to record from identified interneurons using either photo-tagging or GRIN-lens techniques, both of which will take several months to be implemented. Likewise, properly controlled chemogenetic manipulations of specific interneuron populations in vivo will require many months of work and substantial controls. We have therefore discussed more thoroughly the correlation between our in vivo and in vitro experiments in the revised manuscript, highlighting the limitations of our dataset.

3) The concentration of α5IA of 100 nm seems not to provide the highest specificity of the drug, having a milder but significant effect in receptors containing the α3 and α1 subunits (Dawson et al., 2005). Considering the general importance that this strong and very specific effect may have for the community, and the sparse literature using this drug in vitro I would appreciate a better characterization of this particular experiment.– Does α5IA changes the kinetics of the GABAergic response?– Are PV-mediated IPSCs affected by this concentration of α5IA?

The reviewer raises a good concern. However, we are confident that we are blocking α5-GABA_A_Rs selectively. Indeed, in Table 2 of Sternfeld et al., 2004, compound 16 corresponding to α5IA shows -29%, -4% and +4% efficacy at α5, α1 and α3-expressing GABA_A_Rs respectively. In addition, Figure 3 of Dawson et al. indicates that α5-IA at 100nM has less than 10% effect with recombinant α3-expressing GABA_A_Rs and around 15% effect for α1-containing GABA_A_Rs while the effect was above 40% for α5-expressing GABA_A_Rs. Moreover, PV-mediated IPSCs were not affected by this concentration of α5IA in control mice.

To assess a possible effect of the drug on the kinetics of synaptic responses we quantified rise and decay time constants (taus) in the presence and absence of α5IA in the bath. The drug did not affect uIPSC kinetics in both genotypes. Representative uIPSCs traces and population are shown in Author response image 2, and indicated in the revised manuscript (Results section).

– Does the α5IA dependent response shows voltage-dependency as has been reported (Schultz et al., 2019)

We did not test voltage-dependency of the α5IA, as this drug was already characterized at dendritic inhibitory synapses (e.g. Schulz et al., 2018; Ali and Thomson, 2008). The experiments with this drug were not central to the core findings of our manuscript (namely inhibitory circuit-specific alterations in Ts mice). In general, however, voltage-dependency on dendrite-targeting synaptic responses can be problematic due to known space-clamp issues.